# Large scale plasma proteomics identifies novel proteins and protein networks associated with heart failure development

Amil M. Shah [1,2] ✉, Peder L. Myhre[3], Victoria Arthur [1,2], Pranav Dorbala[2], Humaira Rasheed [3,4,5], Leo F. Buckley[6], Brian Claggett[2], Guning Liu[7], Jianzhong Ma[7], Ngoc Quynh Nguyen[7], Kunihiro Matsushita[8], Chiadi Ndumele[8], Adrienne Tin [9], Kristian Hveem[5], Christian Jonasson [5], Håvard Dalen[10,11,12], Eric Boerwinkle[7], Ron C. Hoogeveen[13], Christie Ballantyne [13], Josef Coresh [14], Torbjørn Omland[3] & Bing Yu [7]

Heart failure (HF) causes substantial morbidity and mortality but its pathobiology is incompletely understood. The proteome is a promising intermediate phenotype for discovery of novel mechanisms. We measured 4877 plasma proteins in 13,900 HF-free individuals across three analysis sets with diverse age, geography, and HF ascertainment to identify circulating proteins and protein networks associated with HF development. Parallel analyses in Atherosclerosis Risk in Communities study participants in mid-life and late-life and in Trøndelag Health Study participants identified 37 proteins consistently associated with incident HF independent of traditional risk factors. Mendelian randomization supported causal effects of 10 on HF, HF risk factors, or left ventricular size and function, including matricellular (e.g. SPON1, MFAP4), senescence-associated (FSTL3, IGFBP7), and inflammatory (SVEP1, CCL15, ITIH3) proteins. Protein co-regulation network analyses identified 5 modules associated with HF risk, two of which were influenced by genetic variants that implicated *trans* hotspots within the *VTN* and *CFH* genes.

Heart failure (HF) is a multisystem disorder that affects 5.7 million Americans, costs $30.7 billion annually, and is associated with a 50% 5-year mortality[1]. Although neurohormonal activation is an established biologic pathway underlying HF, much remains unknown regarding HF pathophysiology especially when occurring in the absence of antecedent myocardial infarction or with preserved left ventricular (LV) ejection fraction (HFpEF)[2]. Despite the growing burden of HF, there has been limited progress in leveraging deep molecular phenotyping ('-omics' technologies), such as genomics, to better understand disease mechanisms and biologic sub-phenotypes for precision medicine.

[1]Division of Cardiology, University of Texas Southwestern Medical Center, Dallas, TX, USA. [2]Division of Cardiovascular Medicine, Brigham and Women's Hospital, Boston, MA, USA. [3]Akershus University Hospital and K.G. Jebsen Center for Cardiac Biomarkers, University of Oslo, Oslo, Norway. [4]K.G. Jebsen Center for Genetic Epidemiology, Department of Public Health and Nursing, NTNU, Norwegian University of Science and Technology, Trondheim, Norway. [5]Department of Public Health and Nursing, HUNT Research Center, Norwegian University of Science and Technology, Trondheim, Norway. [6]Department of Pharmacy, Brigham and Women's Hospital, Boston, MA, USA. [7]Department of Epidemiology, Human Genetics, and Environmental Sciences, University of Texas Health Sciences Center at Houston, Houston, TX, USA. [8]Department of Epidemiology, Johns Hopkins Bloomberg School of Public Health, Baltimore, MD, USA. [9]University of Mississippi Medical Center, Jackson, MS, USA. [10]Department of Circulation and Medical Imaging, Norwegian University of Science and Technology, Trondheim, Norway. [11]Clinic of Cardiology, St Olavs University Hospital, Trondheim, Norway. [12]Department of Internal Medicine, Levanger Hospital, Levanger, Norway. [13]Division of Cardiology, Baylor College of Medicine, Houston, TX, USA. [14]Departments of Medicine and Population Health, NYU Langone Health, New York, NY, USA. ✉e-mail: Amil.Shah@utsouthwestern.edu

As proteins are the effectors of genes and their circulating levels are frequently influenced by genetic variation, the proteome is a promising intermediate phenotype for discovery of novel mechanisms underlying HF development. Parabiosis experiments demonstrate important effects of circulating factors on cardiac structure and function[3], highlighting the potential of circulating proteins to influence HF pathobiology. Furthermore, circulating proteins are common targets of pharmacologic therapies[4]. High throughput quantitation of nearly 5000 proteins using aptamer affinity technology therefore holds promise to deepen understanding of HF biology. Prior proteomic studies from the Framingham Heart, Jackson Heart, Malmo, PIVUS, ULSAM, and HOMAGE studies have identified novel markers of HF risk but have been relatively limited in proteome coverage and

sample size[5–10], with sparse data regarding protein networks and potential causality of identified associations. We leveraged robust clinical phenotyping and prospective event adjudication available in two large prospective cohorts with diversity of age, geographic locations, and HF ascertainment to identify proteins robustly and reproducibility associated with HF across these sources of heterogeneity. We evaluated the associations of individual proteins with HF risk and used the correlation structure between proteins to study the associations of protein networks with HF risk. We used data on common and low frequency genetic variants to identify proteins and networks with potential causal associations with HF development (Fig. 1).

In this work, we performed a parallel analysis of the relationship of 4877 aptamers (4697 unique proteins) measured using modified

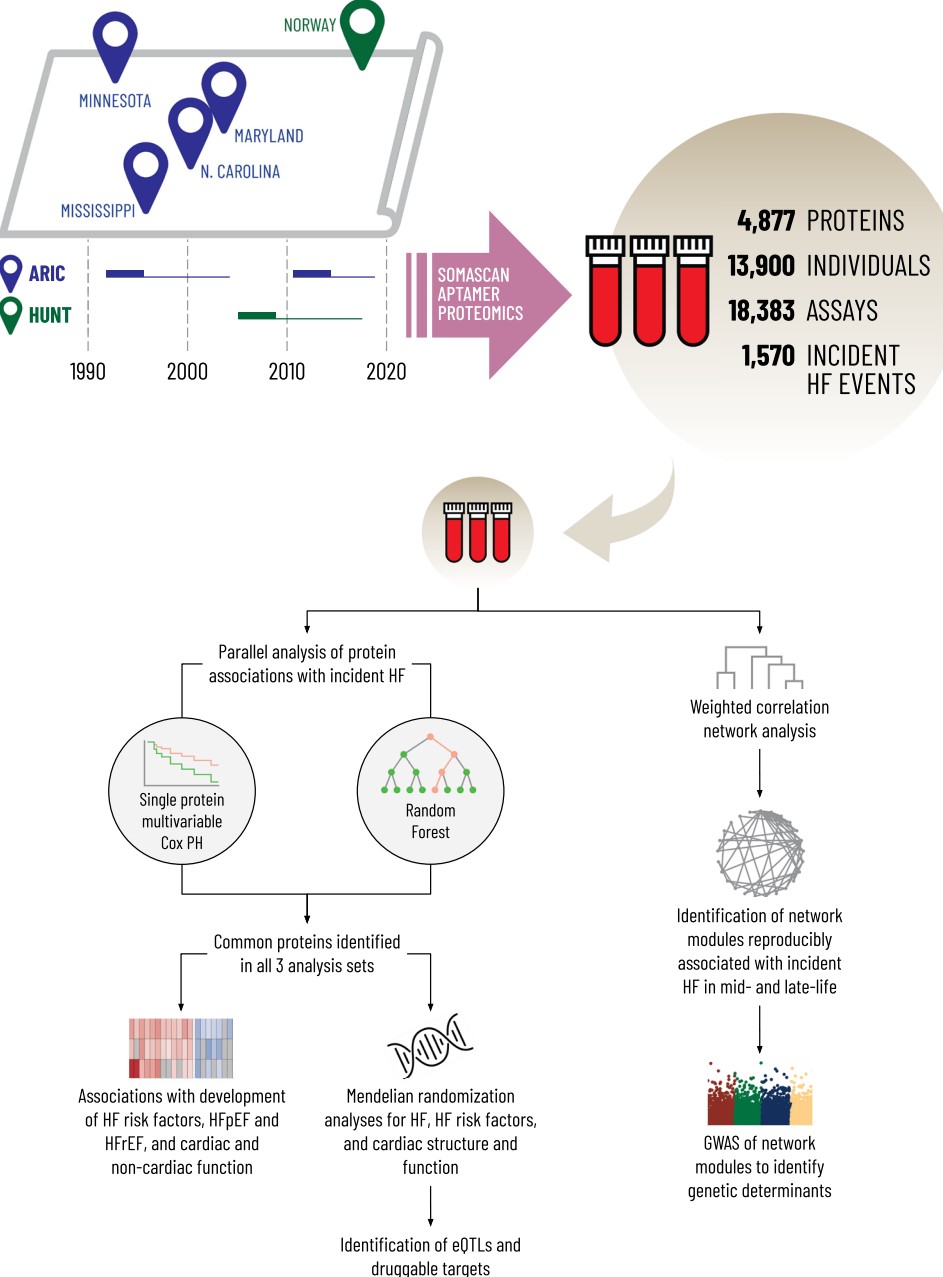

**Fig. 1 | Schematic overview of study design.** The three analysis sets are: ARIC Visit 3 (1993-1995; age 60 ± 5 years, 54% women, 21% Black race), ARIC Visit 5 (2011-2013; age 75 ± 5 years, 58% women, 17% Black race), and HUNT cycle 3 (2006-2008; age 65 ± 10 years, 39% women, 0% Black race). For Mendelian randomization analysis,

protein quantitative trait loci (pQTLs) were obtained from the INTERVAL, AGES, and Fenland studies as instrumental variables (IV). Summary statistics for heart failure were from the HERMES study and summary statistics for cardiac structure and function were from UK Biobank.

**Table 1 | Baseline clinical characteristics and follow-up events for participants included in each analysis set: ARIC mid-life baseline (Visit 3), ARIC late-life baseline (Visit 5), and HUNT**

| | ARIC Visit 3 (1993-1995) | ARIC Visit 5 (2011-2013) | HUNT (2006–2008) |
|---|---|---|---|
| N | 10,638 | 4483 | 3262 |
| Age (years) | 60 ± 6 | 75 ± 5 | 65 ± 10 |
| Black race | 2201 (20%) | 763 (17%) | 0 |
| Male | 4886 (46%) | 1861 (42%) | 1975 (61%) |
| Hypertension | 4839 (45%) | 3622 (81%) | 1352 (41%) |
| Smoking | 6279 (59%) | 2593 (58%) | 2145 (66%) |
| Diabetes | 1936 (18%) | 1560 (35%) | 307 (9%) |
| BMI (kg/m$^2$) | 28 ± 5 | 28 ± 5 | 28 ± 4 |
| eGFR (ml/min/ 1.73 m$^2$) | 86 ± 15 | 72 ± 17 | 88 ± 16 |
| Coronary disease | 673 (6%) | 662 (15%) | 566 (17%) |
| Atrial fibrillation | 112 (1%) | 232 (5%) | 208 (6%) |
| Incident HF events | 822 | 408 | 340 |
| F/U Time (Years) | 10.0 [10.0, 10.0] | 7.2 [5.6, 7.8] | 10.0 [9.2, 10.6] |
| Event Rate | 0.8 [0.8, 0.9] | 1.4 [1.3, 1.5] | 1.2 [1.1,1.3] |

Incident HF events post-ARIC Visit 3 and post-ARIC Visit 5 were not overlapping. Continuous variables are provided as n ± standard deviation or n [25th percentile limit, 75th percentile limit] as appropriate.

aptamer technology (SomaScan v4) with incident HF in Atherosclerosis Risk in Communities (ARIC) study participants in mid-life ($n$ = 10,638; age 60 ± 5 years) and late-life ($n$ = 4483; age 75 ± 5 years) and in Trøndelag Health Study (HUNT) participants ($n$ = 3262; age 65 ± 10 years). We identify 37 proteins (represented by 38 aptamers) significantly associated with incident HF independent of traditional HF risk factors in all three study samples. These proteins demonstrate largely consistent associations with incident HFpEF and HFrEF, but differential associations with risk of developing key HF risk factors from mid- to late-life and with cardiac and non-cardiac physiologic measures in late-life. Potential causal associations with HF, alterations in LV structure or function, or clinical HF risk factors were observed for 10 of these 37 proteins by two-sample Mendelian randomization (MR) analysis (one for HF; seven for LV structure or function; seven for clinical HF risk factors), 8 of which were annotated as druggable targets. Analysis of protein co-regulation networks using weighted co-expression analysis identified five protein modules reproducibly associated with HF risk in mid- and late-life, three of which are influenced by *cis*- and *trans*-acting genetic variants. Additional systems-level analyses of HF-associated proteins provide further insight into relevant mechanistic and regulatory pathways for HF risk.

## Results

### Characteristics of cohorts

A total of 10,638 HF-free ARIC participants at study Visit 3 (1993-1995; age 60 ± 5 years, 54% women, 21% Black race; Table 1) were included in the ARIC mid-life baseline analysis, and experienced 822 incident HF events over a 10-year follow-up. ARIC late-life baseline analysis included 4483 HF-free ARIC participants at study Visit 5 (2011-2013; age 75 ± 5 years, 58% women, 17% Black race), who experienced 408 incident HF events over a median follow-up of 7 [IQR 6, 8] years. The HUNT cohort consisted of 3262 participants from the third study cycle (2006-2008; age 65 ± 10 years, 39% women, 0% Black race), with 340 incident HF events over a median follow-up of 10 [IQR 9, 11] years. Detailed description of participant characteristics, inclusion and exclusion criteria, and participant flow for each analysis set are provided in Supplementary Figs. 1–3. The SomaScan assay consists of 5284 aptamers, 4877 (91%) of which

passed ARIC quality control assessments (see Methods) and were used in this analysis.

### Association of Protein Levels with Incident Heart Failure

In uniprotein multivariable analysis adjusting for demographics and clinical risk factors, associations with incident HF were observed for 948 proteins at an FDR $p < 0.05$ and 283 proteins at Bonferroni significance ($1×10^{-5}$) in the ARIC mid-life analysis; 558 proteins at an FDR $p < 0.05$ and 141 proteins at Bonferroni significance in the ARIC late-life analysis; and 52 proteins at an FDR $p < 0.05$ and 17 proteins at Bonferroni significance in HUNT (Fig. 2a). The direction and magnitude of effect of these proteins with incident HF were generally consistent across analysis sets (Fig. 2b). As a complementary approach to the main parallel analyses, we conducted meta-analysis using the ARIC visit 3, ARIC visit 5, and HUNT data (see Methods) which identified 294 proteins associated with incident HF at a Bonferroni-corrected level of significance (Supplementary Fig. 4). Most of the candidate proteins identified in our parallel analysis were the most strongly associated with incident HF in the meta-analysis. Overrepresentation pathway analysis using these 294 proteins as input identified 3 overrepresented pathways at FDR < 0.05: Hepatic Fibrosis / Hepatic Stellate Cell Activation, LXR/RXR Activation, and Inhibition of Matrix Metalloproteases (Supplementary Data 1).

Thirty-three proteins were associated with incident HF at an FDR < 0.05 in all 3 analysis sets, of which 11 were associated at Bonferroni significance ($p < 1×10^{-5}$) in all three analysis sets. We employed random survival forest analysis as a complementary feature selection approach that simultaneously considered all proteins and did not assume a linear protein level – outcome association (see Methods). Random survival forest analysis, performed in parallel in each analysis set, identified 17 proteins retained in all three analysis sets. Eleven were also identified by the uniprotein multivariable Cox regression analysis while six were newly identified by the random survival forest analysis (Fig. 2c).

The resulting 39 HF-associated aptamers (17 identified through both Cox models and random forest analysis, 16 identified through Cox models alone, 6 identified through random forest analysis alone) demonstrated consistent associations with incident HF across the 3 analysis sets, with higher levels of 30 associated with higher HF risk and higher levels of 9 associated with lower HF risk (Fig. 2d). These 39 aptamers represented 37 unique proteins, one of which was NT-proBNP (Supplementary Fig. 5). Of these 37 proteins, we identified *cis*-pQTLs for all but 4, supporting aptamer specificity (Supplementary Data 2)[11]. We used an orthogonal method to validate aptamer specificity (Olink Explore 3072 proximity extension assay [$n$ = 27], targeted ELISAs [$n$ = 2], or electrochemiluminescence sandwich immunoassay [$n$ = 1]) using plasma from a subset of 113 participants (Supplementary Fig. 6). No validation was available for 7/37 proteins: TAGLN, TREM1, CACNA2O3, FBLN5, CLIP2, CELA1, PTPRD. Correlation was good (>0.70) for the majority of proteins (18/30) assessed, moderate (0.40-0.70) for 8/30, and poor (<0.40) for only 4 (FSTL1, APOF, SVEP1, ATP1B1).

Based on protein annotations in the Human Protein Atlas (https://www.proteinatlas.org)[12], 78% were annotated as secreted, 14% as membrane bound, and 8% as exclusively intracellular (Supplementary Data 3). Based on data from the Genotype-Tissue Expression (GTEx) project[13], tissue expression varied between proteins with a subset demonstrating robust expression in LV and LA appendage tissue (Supplementary Fig. 7). Given the established associations of BNP with HF, HF risk factors, and cardiac function, further analyses did not include BNP aptamers.

### Associations of Candidate Proteins with Development of HF Risk Factors

Following the ARIC mid-life baseline, 6590 participants developed incident hypertension, 3094 incident diabetes, 959 incident chronic

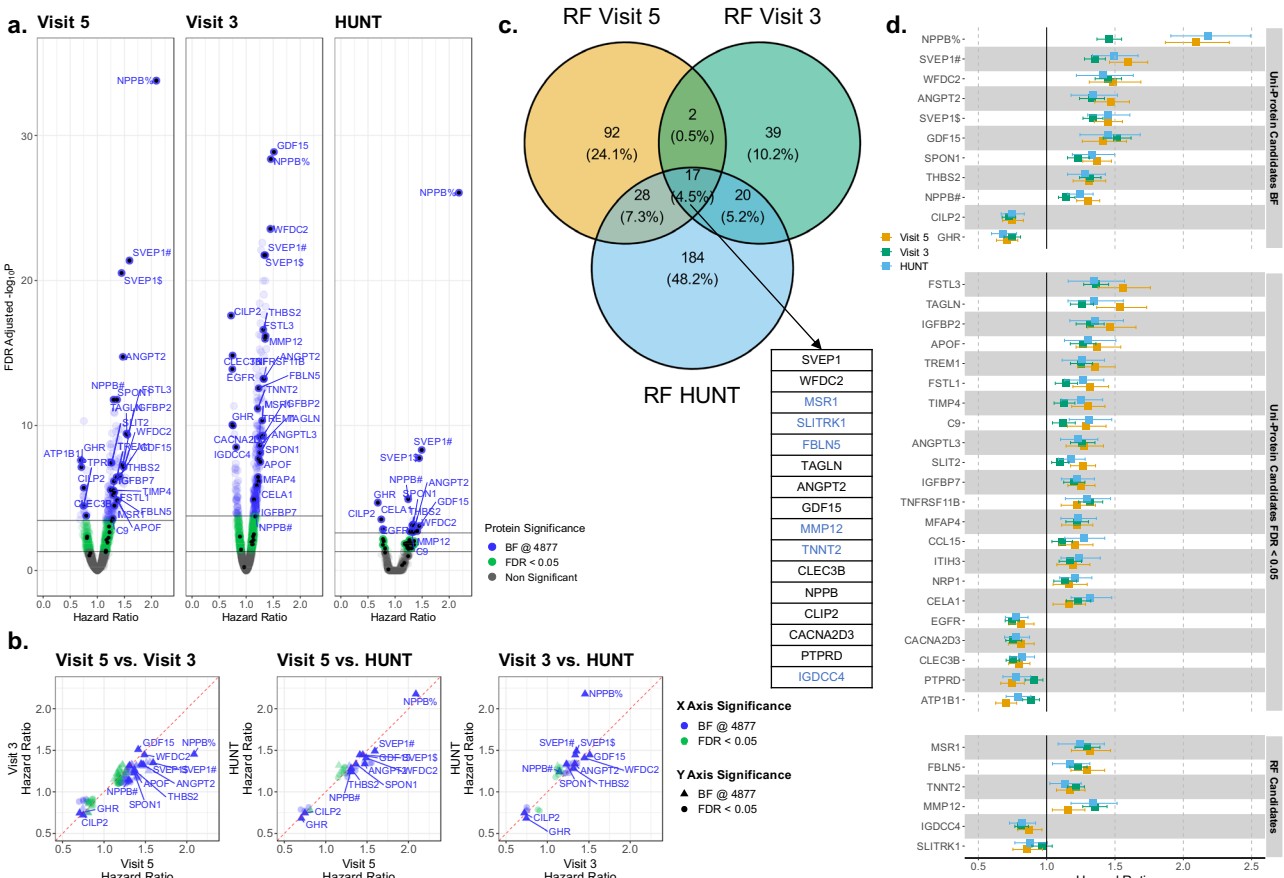

**Fig. 2 | Association of individual proteomic measures with risk of developing HF.** Hazard ratios are based on single protein, multivariable Cox proportional hazard models adjusting for age, BMI, eGFR by CKD-EPI, race, sex, current smoking status, prevalent CAD, prevalent DM, prevalent AF, and prevalent hypertension. **a** Volcano plots of the associations of individual plasma proteins with incident HF in ARIC late-life (Visit 5), ARIC mid-life (Visit 3), and HUNT in models adjusted for demographics and clinical HF risk factors. Green – significant at FDR $p < 0.05$, Blue –Bonferroni (BF) significance ($p < 1×10^{-5}$). **b** Hazard ratio-Hazard ratio plots demonstrating consistency of associations of proteins with incident HF identified in Panel A across analysis sets. Green – significant at FDR $p < 0.05$, Blue –Bonferroni significance ($p < 1×10^{-5}$) in X-axis analysis set; Circle – significant at FDR $p < 0.05$, Triangle –Bonferroni significance ($p < 1×10^{-5}$) in Y-axis analysis set. **c** Random forest (RF) analysis. Venn diagram demonstrating the number of proteins retained by RF analysis [see Methods] in each analysis set, and the overlap between analysis sets. Orange – ARIC late-life baseline (Visit 5), Green – ARIC mid-life baseline (Visit 3), Blue – HUNT. Table shows 16 proteins retained in parallel random forest analysis performed in each analysis set. Light blue indicates proteins not significant at FDR $p < 0.05$ in all three analysis sets in single protein Cox regression models. **d** Forest plot of hazard ratios with 95% confidence intervals for associations with incident HF for proteins associated with HF at FDR $p < 0.05$ or Bonferroni significance in all three analysis sets or retained in random forest analysis in all three analysis sets. Estimates to the left of the horizontal line are associated with a lower risk of incident HF while those to the right are associated with higher risk. Orange – ARIC late-life baseline (Visit 5, $n = 4483$), Green – ARIC mid-life baseline (Visit 3, $n = 10,638$), Blue – HUNT ($n = 3262$). Source data are provided as a Source Data file.

kidney disease (CKD), 2006 incident coronary heart disease (CHD), and 2409 incident atrial fibrillation (AF). The 37 key HF-associated proteins were most consistently associated with risk of incident atrial fibrillation (Fig. 3). Hierarchical clustering identified three protein clusters based on their associations with the development of HF risk factors. One of these clusters (Cluster 2) consisted of 7 proteins, all associated with lower risk of HF, which tended to associate with lower risk of developing several HF risk factors. Cluster 1 consisted of 12 proteins including GDF-15, FSTL3, and IGFBP7, and demonstrated robust associations with higher risk of developing most HF risk factors. The largest cluster (Cluster 3) consisted of 17 proteins, including SVEP1, SPON1, CCL15, and ITIH3. These proteins tended be associated with higher risk of incident atrial fibrillation (except for two proteins associated with lower HF risk, SLITRK1 and PTPRD) and variable associations with incident CKD and/or diabetes.

## Protein Associations with HF Phenotype and Pathophysiologic Measures

Concomitant data on cardiovascular structure and function, and function of non-cardiovascular systems relevant to HF risk

(pulmonary function, body composition, skeletal muscle strength, anemia), were available in the ARIC late-life baseline analysis set for cross-sectional analyses (Fig. 4a). With these measures, three protein clusters were identified using hierarchical clustering. The first cluster of 9 proteins, all associated with lower risk of HF, demonstrated modest associations with cardiac structure and function but more consistent associations with better pulmonary function and lower fat mass. Notable among this cluster was GHR, which associated with lower LV and left atrial (LA) volumes and higher hemoglobin. The two remaining clusters consisted of proteins associated with higher HF risk. Proteins in Cluster 3 included SVEP1, SPON1, FSTL3, IGFBP7, MFAP4, and APOF, and were associated with larger LV and LA size, higher E wave velocity, higher LV filling pressure (greater E/e' ratio and LA volume), and worse LA function (LA reservoir and contraction strains). These proteins also demonstrated robust associations with non-cardiovascular measures including lower fat mass, lower grip strength, lower hemoglobin, and less consistently with worse pulmonary function – all potential extracardiac contributors to HF. Cluster 2 proteins, which included NRP1, CCL15, ANGPTL3, and ITIH3, demonstrated less consistent associations with cardiovascular

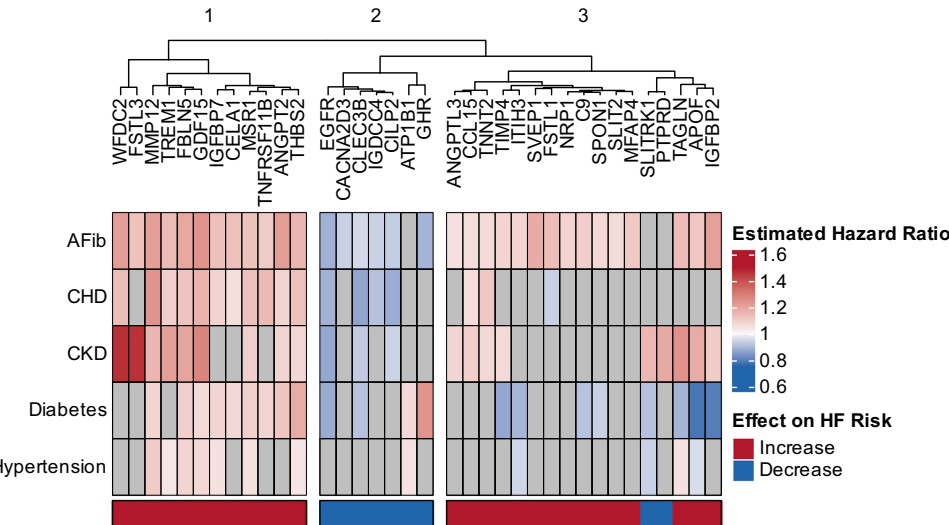

**Fig. 3 | Associations of HF-related proteins with HF-risk factors.** Heatmap showing the associations of HF-associated proteins (Fig. 2d above) assessed at ARIC mid-life baseline (Visit 3) with the risk of incident diseases that are recognized as HF-risk factors post-Visit 3. Color shading indicates hazard ratio as indicated in the scale. Gray indicates non-significant association. Cox proportional hazard models adjusted for age, BMI, eGFR by CKD-EPI, race, sex, current smoking status, prevalent CAD, prevalent DM, prevalent AF, and prevalent hypertension, excluding the outcome variable from the adjustment, with FDR p-value < 0.05 considered significant. Color of the bar below the heatmap signifies the observed association of protein levels with incident HF risk. Red – higher protein level associates with higher risk of incident HF; Blue – higher protein level associates with lower risk of incident HF. Proteins were ordered using hierarchical clustering based on associations with incident HF risk factor development. Source data are provided as a Source Data file.

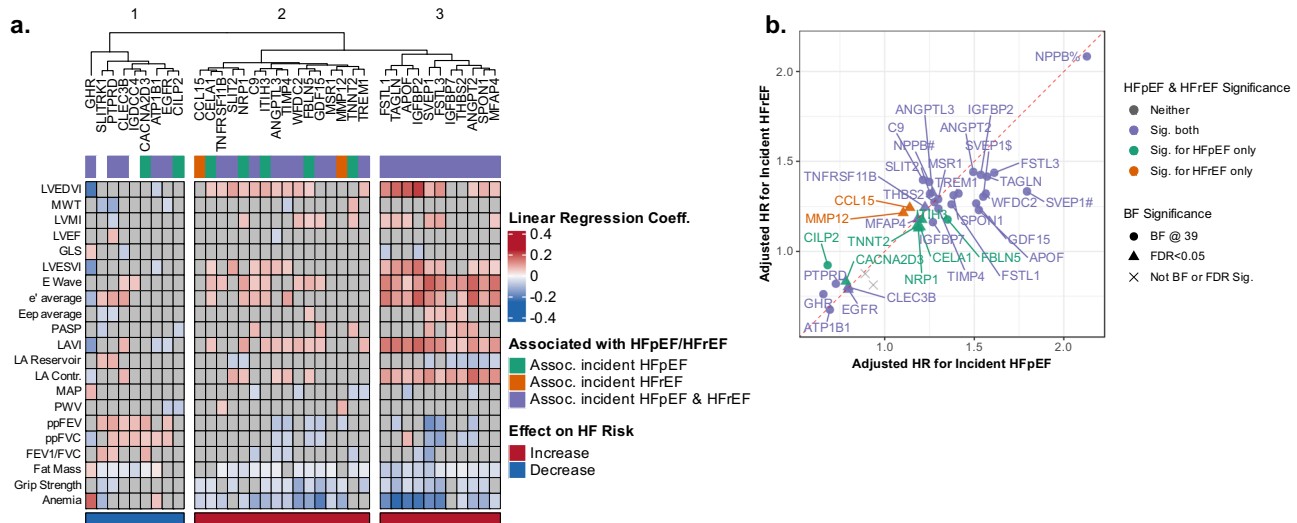

**Fig. 4 | Associations of HF-related proteins with cardiac and non-cardiac function and incident HF phenotype in late-life. a** Heatmap showing the cross-sectional associations of HF-associated protein levels with measures of cardiac structure and function, arterial properties, pulmonary function, fat mass, grip strength, and anemia. Proteins were ordered using hierarchical clustering based on associations with cardiovascular and non-cardiovascular function. Linear regression models adjusted for age, BMI, eGFR by CKD-EPI, race, sex, current smoking status, prevalent CAD, prevalent DM, prevalent AF, and prevalent hypertension, and additionally for heart rate and systolic blood pressure for echocardiographic outcomes. Color shading indicates hazard ratio as indicated in the scale. Gray indicates non-significant association. FDR p-value < 0.05 was considered statistically significant after adjusting for multiple testing. Color of the bar below the heatmap signifies the observed association of protein levels with incident HF risk. Red – higher protein level associates with higher risk of incident HF; Blue – higher protein level associates with lower risk of incident HF. Proteins were ordered using hierarchical clustering based on associations with cardiovascular and non-cardiovascular function. **b** Hazard ratio-Hazard ratio plot demonstrating consistency of associations of candidate proteins with incident HFpEF (X axis) and HFrEF (Y axis) in the ARIC late-life analysis set (Visit 5). Hazard ratios (HRs) are based on single protein, multivariable Cox proportional hazard models adjusting for age, BMI, eGFR by CKD-EPI, race, sex, current smoking status, prevalent CAD, prevalent DM, prevalent AF, and prevalent hypertension. Color indicates outcomes for which statistically significant associations were observed (Orange – HFrEF, Green – HFpEF, Purple – Both, Gray – neither), which shape indicates strength of association (Triangle – FDR < 0.05, Circle – Bonferroni-corrected (BF) significance level). Source data are provided as a Source Data file.

measures compared to Cluster 3, but similar associations with non-cardiovascular measures.

Incident heart failure phenotype (HFpEF and HF with reduced LVEF (HFrEF)) was available in the ARIC late-life baseline analysis set (193 incident HFpEF, 157 incident HFrEF). Of the 37 key HF-associated proteins, effect estimates were generally similar for incident HFpEF and incident HFrEF, with all but 8 demonstrating significant associations with both (Fig. 4b).

## Mendelian Randomization Causally Implicates Proteins

For the 37 top candidate proteins that were significantly associated with incident HF, a two sample Mendelian randomization (MR) approach was employed to assess potential causal relations of those proteins with HF, with key HF risk factors, or with alterations in LV size and function using protein quantitative trait loci (pQTLs) from the INTERVAL, AGES, and Fenland studies as instrument variables (IV) (see Methods)[14–16]. A single *trans*-pQTL for SVEP1 (rs687621 in the *ABO* gene) demonstrated MR association with HF (Fig. 5a; Supplementary Data 4). We observed MR evidence for associations between genetically regulated levels of 7 proteins and HF risk factors (Fig. 5a) including: SPON1, CCL15, and ITIH3 with hypertension; SVEP1 and SPON1 with atrial fibrillation; FSTL3 and NRP1 with diabetes; and APOF, CCL15, ITIH3, and NRP1 with CHD. We did not observe MR evidence for significant causal associations with CKD. Two sample MR was also performed using summary statistics for LVEDV, LVESV, and LVEF assessed by cardiac MRI in UK Biobank ($n = 36{,}041$) to assess potential causal associations with cardiac remodeling and dysfunction. Genetically regulated levels of 7 proteins were associated with LV measures (Fig. 5b). Genetically higher levels of SPON1 associated with higher LVEF, lower LVEDV, and lower LVESV. Additional significant MR associations included: IGFBP7 with LVESV and LVEF; CCL15 with LVEDV and LVESV; FSTL3, ANGPTL3, and NRP1 with LVEDV; and MFAP4 with LVEF. Analyses using only *cis*-acting pQTLs demonstrated generally similar findings, although MR association of SVEP1 with HF was no longer observed (Supplementary Fig. 8). Replication of significant single SNP MR results in distinct pQTL datasets was available for 5 protein-outcome associations, all of which demonstrated consistent results (Supplementary Data 5). Backward MR for significant associations, which used instrumental variables for HF, HF risk factors, or cardiac structure and function as exposure and instrumental variables for proteins as the outcome, did not detect evidence of potential reverse causality (Supplementary Data 6). Colocalization analysis demonstrated evidence of colocalization for the majority of MR significant associations (19 of 34; Supplementary Data 7). We observed evidence of colocalization for observed MR associations for SPON1, CCL15, and FSTL3 with multiple outcomes; for SVEP1 with HF; and for NRP1 with diabetes and CHD. Power was inadequate for colocalization for ANGPTL3 and MFAP4. For the observed associations for NPPB, ITIH3, and IGFBP7 with multiple outcomes and for the association of SVEP1 with atrial fibrillation, our findings were negative for colocalization, suggesting two different causal variants for protein level and outcome.

Supportive of our MR findings, two SPON1 pQTLs (rs10832169, rs1969539) were also annotated as eQTLs in GTEx and were associated with altered SPON1 expression in the LV and LA appendage, in addition to liver, visceral adipose, skeletal muscle, and tibial artery (Supplementary Fig. 9). Additionally, a pQTL for MFAP4 (rs139356332) was associated with altered protein expression in the LA appendage while a pQTL for ITIH3 (rs2535629) was associated with altered protein expression in whole blood. Based on data from the druggable genome database[17], there are already existing agents (antibodies) targeting NRP1 and ANGPTL3, both with MR evidence of potential causal associations with LV size. An additional 6 proteins with MR evidence of potential causal associations with LV measures and/or HF risk factors were annotated as druggable (CCL15, FSTL3, IGFBP7, ITIH3, MFAP4, APOF; Supplementary Data 8)[17].

Finally, we performed consensus clustering based on levels of the 10 proteins with MR support for causal associations with HF, HF risk factors, or cardiac structure/function (see Methods). Application of consensus clustering identified three clusters based on protein profiles in the ARIC late-life analysis set (Fig. 6; Supplementary Fig. 10). Relative to the largest cluster, one cluster consisted of participants with higher levels of multiple proteins and were at the highest risk of incident HF. Participants in another cluster demonstrated lower levels of multiple proteins and were at the lowest risk of developing HF.

## Network Analysis Identifies Protein Modules Relevant to Heart Failure Risk

By applying weighted correlation network analysis (WGCNA) to the ARIC late-life analysis set (see Methods)[18], we identified 28 protein modules, 6 of which were associated with incident HF at a Bonferroni adjusted level of significance (Fig. 7a, b). When applied to the ARIC mid-life analysis set, 5 of these modules (brown, pink, light green, white and salmon) were also associated with incident HF in mid-life at a Bonferroni adjusted level of significance (Fig. 7c). The largest of these, the brown module, consisted of 409 proteins and constituted 2 large sub-modules (Fig. 7d), both of which demonstrated robust associations with incident HF in both mid- and late-life (Supplementary Data 9). Submodule 1 consisted of 312 proteins, including 17 of the top HF-associated proteins identified in the single-protein analysis, and was enriched for several canonical pathways including Ephrin receptor signaling, STAT3 pathway, atherosclerosis signaling, and PI3K/AKT signaling relevant to cardiac hypertrophy and remodeling, among others (Supplementary Data 10). GWAS of this sub-module, summarized using a module eigengene, from both the mid- and late-life analysis sets identified 94 associated variants in the *CFH* gene on chromosome 1. LD-based clumping identified two clumps: Clump 1 included a missense variant in *CFH* (rs1061170) while the independent SNP for Clump 2 was rs424535, which was nominally associated with HF in the HERMES consortium ($p = 0.027$). Both SNPs demonstrated *trans* effects on a large number of proteins across several modules beyond the Brown submodule. Submodule 2 consisted of 97 proteins, including 4 of the top HF-associated proteins from the single-protein analysis, and was enriched for canonical pathways related to dendritic cell maturation (Supplementary Data 10).

The Pink module consisted of 30 proteins, none of which were identified as top HF-associated proteins in our single protein analysis. Greater Pink module eigenvalues associated with lower HF risk in both mid- and late-life analysis sets. GWAS identified 255 SNPs associated with the Pink module in both analysis sets, all of which localized to a single region on Chromosome 17 (Fig. 7e). There were 34 independent SNPs identified using LD-based clumping, which influenced levels of 18 out of 30 module proteins. These included SNP rs704, a missense variant in the *VTN* gene, which was associated with levels of all 18 proteins. Rs704 SNP demonstrated *trans* effects on many proteins, but the pink module had the largest proportion of proteins influenced by this SNP.

The light green module consisted of 19 proteins (Fig. 7f) and associated with lower HF risk in both ARIC analysis sets. GWAS identified two SNPs (rs1035849, rs1017301) in high LD on chromosome 12 related to gene *PZP*, and of the 19 module proteins, these SNPs only influenced PZP levels. The salmon module consisted of 23 proteins, 2 of which were identified as top HF-associated proteins in our single protein analysis, and enriched for acute phase response signaling, complement system, lipid metabolism, and senescence pathways (Fig. 7g; Supplementary Data 11). The White module consisted of 12 proteins. No consistent genetic determinants for the white and salmon modules were identified.

## Discussion

We measured 4877 plasma proteins in 13,900 HF-free individuals across three analysis sets comprising 18,383 assessments with diverse age, geography, and HF ascertainment to identify circulating proteins and protein networks robustly associated with HF development. Using multivariable Cox PH regression and a complementary random forest analysis (a supervised machine learning approach), we identified 37 unique proteins that reproducibly associated with HF risk in all three analysis sets independent of traditional HF risk factors. While preclinical data supports the involvement of most of these in HF development, circulating levels of only 8 have previously been associated with risk of incident HF and 10 with risk of adverse outcomes among

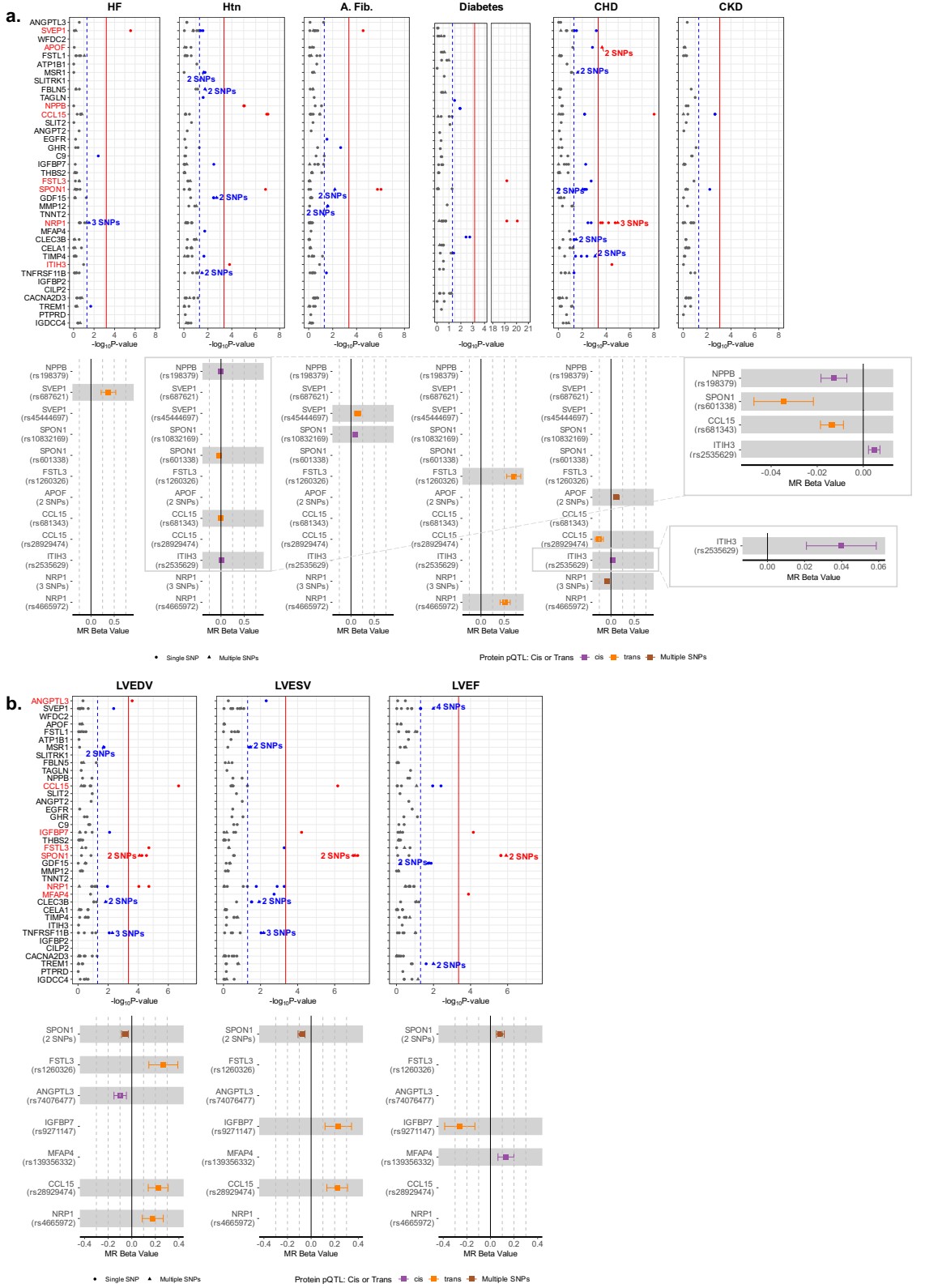

patients with prevalent HF (excluding NT-proBNP and troponin T; Supplementary Data 12).These proteins tended to associate with both incident HFpEF and HFrEF but demonstrated differential associations with the development of HF risk factors and with alterations in cardiac function cross-sectionally. We found MR support for potentially causal associations of 10 proteins with HF, HF risk factors, or LV size and function, 8 of which were druggable targets. Protein co-regulation

network analyses identified 5 protein modules associated with HF risk, three of which contained no proteins identified in our candidate protein analysis. Three modules were influenced by *cis*- and *trans*-acting genetic variants and implicated two *trans* hotspots within the Vitronectin (VTN) and Complement Factor H (CFH) genes.

Compared to prior population-based studies relating large panels of circulating proteins with risk of incident HF, our study is unique in

**Fig. 5 | Results of two-sample Mendelian randomization (MR) analyses.**
**a** Manhattan plots of MR analyses of protein candidates with HF and HF risk factors. pQTLs are single SNP markers unless otherwise labeled. Hatched line indicates nominal significance ($p < 0.05$). Solid red line indicates significance after Bonferroni multiple testing correction. Gray pQTLs – non-significant, Blue – nominally significant, Red – significant after multiple testing correction. Forest plots demonstrate direction and magnitude of effect (MR beta estimate and 95% confidence interval) of genetically higher protein levels for significant pQTLs. Purple – *Cis* pQTL, Orange – *Trans* pQTL, Brown multi-SNP pQTL comprising both *cis* and *trans* SNPs. pQTLs were obtained from the INTERVAL ($n = 3301$), AGES ($n = 5368$), and Fenland ($n = 10,708$) studies. The summary statistics for HF were obtained from the

HERMES consortium ($n = 977,323$). Summary statistics for atrial fibrillation were obtained from a GWAS meta-analysis of 6 studies ($n = 1,030,836$), for CHD were from UK Biobank and replicated using CARDIoGRAMplusC4D data ($n = 296,525$), for CKD were from a 43 study GWAS meta-analysis ($n = 117,165$), for DM were from a GWAS meta-analysis of 3 studies ($n = 655,666$), and for hypertension were from a UK Biobank GWAS ($n = 463,010$). **b** Manhattan plots of Mendelian randomization analyses of protein candidates with measures of left ventricular size and function (LVEDV, LVESV, LVEF). Forest plots show causal estimates with 95% confidence intervals. Summary statistics for LVEDV, LVESV and LVEF were obtained from UK Biobank ($n = 36,041$). See Methods. Source data are provided as a Source Data file.

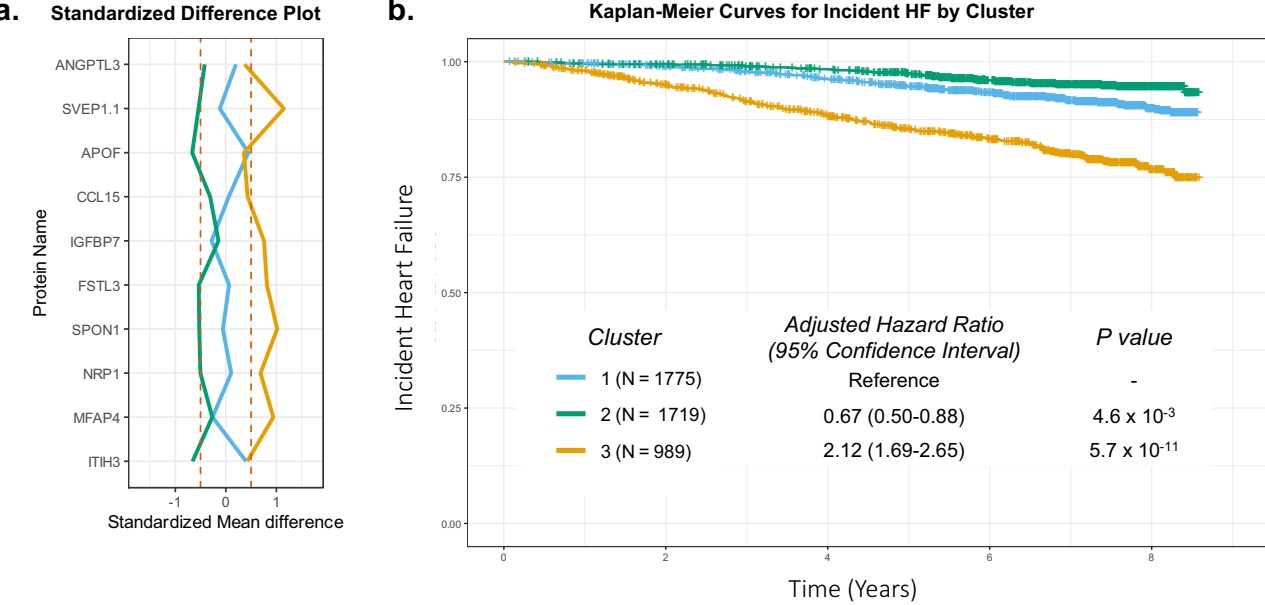

**Fig. 6 | ARIC late-life participant clusters based on the 10 proteins with significant associations in MR analyses derived using consensus clustering algorithm. a** Plot of the mean standard difference in the 10 proteins values in each cluster versus the overall sample. Hatched red lines indicate standard differences of 0.5 and −0.5. **b** Kaplan-Meier curves for incident HF after the ARIC late-life baseline

by cluster assignment and adjusted hazard ratios for incident HF associated with clusters 2 and 3 with cluster 1 as reference. The Cox proportional hazards model was adjusted for age, gender, Field Center, hypertension, diabetes, BMI, atrial fibrillation, smoking status, CHD, and eGFR. Source data are provided as a Source Data file.

its parallel analysis design that sought to ensure consistent, statistically robust associations across important sources of heterogeneity in the HF syndrome: age, geographic location, and method of event ascertainment. Plasma collection and storage procedures in HUNT that closely reflect clinical practice conditions further ensure generalizability of our findings. We identified several established markers of HF risk (NT-proBNP, GDF-15)[19,20], confirmed previously described associations of other proteins with incident HF (e.g. SPON1[8–10], THBS2[5,7], GHR[6], IGFBP2[5,6], C9[7], osteoprotogerin[8,10], MMP12[8–10]), and identified several proteins not previously associated with incident HF (e.g. SVEP1, FSTL3, APOF, ANGPTL3, IGFBP7, CCL15, and ITIH3 among others). The proteins we identified were predominantly annotated as secreted, and implicated in extracellular matrix remodeling, fibrosis, and inflammation. The proteins were largely consistent in their associations with risk of developing both HFpEF and HFrEF. While possibly related to our use of any HF as an outcome, our findings highlight the shared biologic pathways underlying both HF phenotypes and are consistent with prior findings of a high proportion of proteins correlations shared between prevalent HFpEF and HFrEF[21].

In contrast, only a subset of the 37 proteins were cross-sectionally associated with cardiac structure and function – primarily LV and LA size and diastolic function. These findings support the importance of extra-cardiac dysfunction in HF pathophysiology. In particular, consistent associations of HF-risk promoting proteins with lower fat mass,

grip strength, and anemia suggest the importance of frailty[22,23]. That proteins associated with lower HF risk demonstrated consistent associations with better pulmonary function highlights the relevance of cardiopulmonary interactions to HF development[24]. Similarly, while a subset of proteins demonstrated consistent associations with development of several HF risk factors, the most consistent association across the 37 proteins was with incident atrial fibrillation, with relatively few proteins associated with incident CHD or hypertension. These findings support the recognized intimate relationship between atrial fibrillation and HF, particularly HFpEF[25], while the lack of association of several proteins with incident CHD and hypertension may be partly related to our analytic approach which included a late-life analysis set among whom hypertension is highly prevalent.

Mendelian randomization analyses supported causal associations for 10 of 37 proteins with HF, HF risk factors, or LV structure and function. These included matricellular proteins implicated in modulating the extracellular matrix, senescence-associated proteins, and proteins related to lipid metabolism, CAD, and inflammation. SVEP1 was the only protein with MR suggestion of a causal association with HF and demonstrated the strongest observed association with incident HF in all three analysis sets. SVEP1 is a ligand for integrin alpha9beta1, promoting cellular adhesion in response to pro-inflammatory signaling[26]. It has also been implicated in maintenance of vascular integrity via interaction with ANGPT2, another HF-associated protein

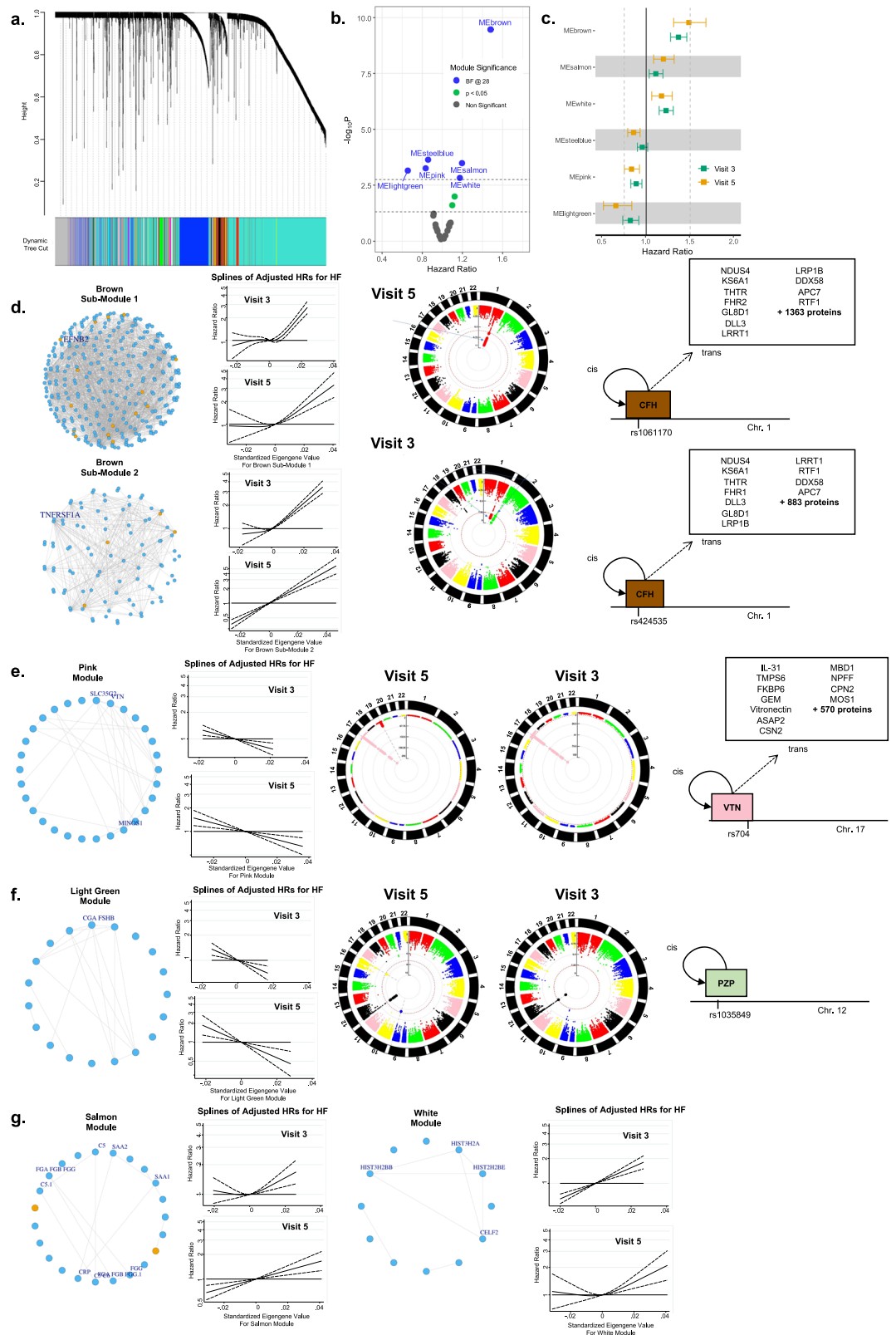

identified in this analysis[27,28]. A low frequency SVEP1 missense variant has previously been associated with an increased risk of CHD, DM, and with higher blood pressure[29]. Genetically higher SVEP1 levels were nominally associated with CHD but also associated with AF in our MR analysis, consistent with our observed association of SVEP1 with incident AF. While SVEP1 has not been previously associated with incident HF, plasma SVEP1 levels are strongly and positively associated with

older age[30] and related disorders like dementia[31], and have recently been described as strongly predictive of outcomes in prevalent HFrEF[32].

MR analysis supported a potential causal association of SPON1 with LV size and function. SPON1 was another protein robustly associated with HF development in all three analysis sets, consistent with prior studies of incident HF and in prevalent HFpEF and HFrEF[8–10,21,33].

**Fig. 7 | Weighted correlation network analysis. a** Hierarchical clustering dendrogram of plasma proteins from the ARIC late-life baseline (Visit 5) using dynamic tree cut identifies 28 protein modules. **b** Volcano plot showing the associations of module eigengene values with incident HF in Cox proportional hazard models adjusted for age, BMI, eGFR by CKD-EPI, race, sex, current smoking status, prevalent CAD, prevalent DM, prevalent AF, and prevalent hypertension. Green – modules significant at FDR $p < 0.05$, Blue – modules significant at Bonferroni-corrected significance level. **c** Forest plot demonstrating hazard ratios (HRs) with 95% confidence intervals for incident HF for HF-associated modules in the ARIC late-life baseline (Visit 5; orange; $n = 4483$) when applied to the ARIC mid-life baseline (Visit 3; green; $n = 10,638$) analysis set. HRs adjustment as in (**b**). **d** Brown module, submodules 1 and 2 network diagram, incidence rate splines for association with incident HF, and GWAS Manhattan plots. Both submodule eigengenes were consistently associated with greater risk of incident HF. Orange network nodes indicate HF-associated proteins identified in candidate analysis (Fig. 2).

GWAS of submodule 1 eigengene value at Visit 3 and Visit 5 identified consistent genetic associations with SNPs in the complement factor H (CFH) gene on chromosome 1. Independent SNPs for two identified LD-based SNP clumps included a missense variant rs1061170 for Clump1 and rs424535 for Clump 2. Both demonstrated *cis* effects on CFH and *trans* effects on many proteins. **e** Pink module. Module eigengene was associated with consistently lower risk of incident HF. GWAS of the eigengene value at Visit 3 and Visit 5 identified consistent genetic associations with 34 independent SNPs in the vitronectin (VTN) gene on chromosome 17, including rs704 which demonstrated *cis* effects on VTN and *trans* effects on many proteins. **f** Light green module. Higher eigengene values were associated with consistently lower risk of developing HF. GWAS identified consistent associations with 2 SNPs in high LD in the pregnancy zone protein (PZP) gene on chromosome 12, which demonstrated *cis* effect on PZP. **g** Salmon and White modules. Both modules demonstrated consistent associations with higher risk of incident HF. Source data are provided as a Source Data file.

SPON1 is an matricellular protein that structurally consists of 6 thrombospondin domains, one spondin domain, and one reelin domain, and has been primarily implicated in dementia and axonal development[34]. Notably, SPON1 is expressed in the LV and LA among other tissues, and the relevant SPON1 pQTLs were also eQTLs in these tissues. Animal models implicate SPON1 in genetically mediated hypertension and in cardiac remodeling due to ischemia-reperfusion[35,36], consistent with our MR results supporting a causal association of SPON1 with hypertension and atrial fibrillation. Another member of the same family, SPON2 (mindin), appears to attenuate cardiac hypertrophy, fibrosis, and dysfunction in response to pressure overload or neurohormonal activation[37,38]. The thrombospondin THBS2 was also robustly associated with incident HF in our analysis. Although we did not observe MR findings of potentially causal association, THSB2 has previously been identified in CAD GWASs[39–42]. Like SPON2, preclinical studies implicate myocardial THBS2 expression in protection from myocardial fibrosis and dysfunction with aging or in response to hypertension[43,44]. Another HF-related matricellular protein with MR findings supporting a causal association with LV function is MFAP4. Similar to SPON2 and THBS2, in pre-clinical models, myocardial MFAP4 expression appears to protect from myocardial fibrosis and hypertrophy in response to pressure overload or neurohormonal activation[45,46]. Furthermore, the relevant MFAP4 pQTL is also an eQTL in left atrial tissue, and plasma MFAP4 levels have previously been associated with atrial fibrillation and degree of atrial fibrosis[47]. Together, these findings highlight the potential importance of matricellular proteins and ECM remodeling to HF development.

Our analysis also implicated proteins known to be associated with the senescence phenotype and age-related cardiac dysfunction as potentially causally associated with LV remodeling and dysfunction. FSTL3 is a downstream regulator of Activin type II receptor signaling, which has been implicated in aging-related muscle wasting[48]. Cardiac myocyte and endothelial cell FSTL3 expression is increased in HF and contributes to paracrine activation of cardiac fibroblasts to promote fibrosis[49–51]. In preclinical models, increasing activin A resulted in cardiac dysfunction through increased ActRII signaling and FSTL3 expression[48]. Similar to GDF15, IGFBP7 is also a component of the senescence associated secretory phenotype[52]. MR analysis supported a causal association of genetically higher circulating IGFBP7 levels with larger LVESV and lower LVEF. Although prior associations of circulated IGFBP7 with incident HF have been limited, higher IGFBP7 levels are associated with worse outcomes among patients with HFpEF and HFrEF[53,54].

Notably, 8 out of the 10 proteins with significant MR findings were annotated as druggable targets by the druggable genome database[17,55], with existing agents targeting 2 proteins (NRP1, ANGPTL3). While NRP1 promotes angiogenesis in malignancy, it is also expressed in the heart and has been implicated in cardiovascular development and maintenance of cardiac function[56]. Mice with NRP1 knocked out in

cardiomyocyte and vascular smooth muscle cells demonstrate decreased survival, cardiac dysfunction, and aggravated ischemia-induced HF[57]. Consistent with this, higher NRP1 levels were associated with greater LVEDV, LVMi, and LAVi in our analysis, and genetically higher NRP1 levels associated with CHD and larger LVEDV. These findings suggest a potential role for anti-NRP1 antibodies, currently being tested in oncology, for HF prevention[58]. ANGPTL3 is expressed in the liver and is an endogenous inhibitor of lipoprotein lipase and endothelial lipase, with loss of function variants associated with lower triglyceride and LDL cholesterol levels and lower odds of CAD[59,60]. Associations with HF have not been previously reported. In our analysis, ANGPTL3 levels were not associated with incident CHD, but MR supported a potential causal association of genetically higher circulating ANGPTL3 levels with lower LVEDV. The potential mechanisms mediating these associations will require further investigation. Importantly, we demonstrate that these 10 proteins with potentially causal associations with HF, cardiac function, or HF risk factors, identify a sizeable subgroup of older persons – approximately one in five – at particularly high risk for developing HF. That most of these proteins are also druggable suggests that this represents a population of at-risk individuals amenable to novel targeted therapies to decrease HF risk.

Single protein-outcome association analyses do not account for the complex correlation structure between circulating proteins. Using weighted gene coexpression network analysis to generate protein networks based on the topological scale-free criteria[18,61,62], we identified five protein modules reproducibly associated with HF development in both mid- and late-life. Two of these modules (Brown, Salmon) contained most of the proteins identified in our single protein analysis while some modules (Pink, Light Green, White) did not contain any, highlighting the complementary information provided by this approach. We observed consistent evidence of genetic regulation in mid and late-life of three of these modules (Brown submodule 1, Pink, Light Green). Genetic associations with the Brown and Pink modules identified known *trans* pQTL 'hotspots' in the *CFH* and *VTN* genes respectively[14–16,61]. Interestingly, these variants in *CFH* and *VTN* are both causally implicated in age-related macular degeneration (AMD)[63,64], an age-related disease that shares many risks factor with HF and is itself associated with heightened risk of HF[65]. AMD and HF also share putative mechanistic pathways, including oxidative stress, inflammation, and mitochondrial dysfunction, suggesting possible shared biology underlying these two distinct late-life conditions[66,67].

While this study has several unique strengths, our findings should be interpreted in the context of the study's limitations. While ascertainment and adjudication of HF varied across analysis sets, this strengthens the generalizability of our findings. We also had limited data regarding the etiology of HF. Data on HF phenotype (HFpEF versus HFrEF) was only available in the ARIC late-life analysis set when the number of incident events were limited. As mechanisms of HF

partially differ by etiology and phenotype, our approach highlights likely shared protein risk factors. We assessed plasma proteins which may not reflect myocardial expression, although some candidate pQTLs are also eQTLs in cardiac issue. However, we believe sampling plasma proteins is appropriate as HF is a systemic disorder influenced by cardiac and extra-cardiac factors, circulating factors are known to influence HF development and pathobiology, and plasma proteins can be readily targeted by certain therapeutics such as antibodies. Further translational studies will be necessary to elucidate the relationships of candidate plasma protein levels with myocardial expression and activity. While the SOMAscan assay provides the broadest coverage of the plasma proteome currently available, it is incomplete and potentially biased in protein sampling. Target specificity of some aptamers may be limited, however we identified *cis* pQTLs for the majority of our candidate proteins. Two-sample MR used GWAS data from HERMES for the HF outcome and UKBB for the CMR outcomes. As some individuals in HERMES and UKBB may have contributed to pQTL data, there is a risk of winner's curse which may result in larger effect size and potentially impact statistical significance. Finally, mid-life and late-life analysis sets both originated from the ARIC cohort and therefore involved overlapping participants at different stages of life, although incident HF events in each analysis set were unique. However, HUNT was a third parallel analysis set and was completely independent of ARIC.

High throughput proteomic profiling in diverse cohorts enables identification of novel protein markers of risk for HF development, and integration with genomic data and annotated bioinformatic databases suggest several may represent targetable HF risk factors. Further studies are necessary to understand the mechanisms by which these proteins influence HF risk.

## Methods

### Study population
The Atherosclerosis Risk in Communities (ARIC) study is a prospective epidemiologic cohort study whose design and methods have been previously described[68]. Between 1987 and 1989, 15,792 middle-aged subjects were enrolled in 4 communities in the United States: Forsyth County, NC, Jackson, MS, suburban Minneapolis, MN, and Washington County, MD. Participants underwent four exam visits between 1987 and 1998, followed by a fifth exam visit between 2011 and 2013. The mid-life baseline analysis set in this analysis used data from ARIC participants attending the third study visit which occurred between 1993-1995 who were free of heart failure (HF) at the time of study visit. The late-life baseline analysis set in this analysis used data from ARIC participants attending the fifth study visit which (2011-2013) who were free of HF at the time of the study visit. The study protocol was approved by institutional review boards at each field center: University of North Carolina at Chapel Hill, Chapel Hill, NC; Wake Forest University, Winston-Salem, NC; Johns Hopkins University, Baltimore, MD; University of Minnesota, Minneapolis, MN; and University of Mississippi Medical Center, Jackson, MS. All participants provided written informed consent at each study visit.

The Trøndelag Health Study (HUNT) Study is a population-based cohort study that collected detailed socio-demographic and clinical information for ~229,000 participants from Trøndelag County in Norway[69]. The current study included 3262 individuals from the third survey (HUNT 3) which enrolled a total of 50,807 participants between 2006–2008, with a follow-up time of 10 years[70]. Proteomic profiling was performed in three cohorts of patients from this study: 1) 971 participants with chronic coronary syndrome (history of myocardial infarction (MI), angiographic evidence of at least 50% stenosis in 1 or more coronary vessels, prior evidence of inducible ischemia by stress testing, or history of coronary revascularization);[71] 2) 1067 participants with an incident primary cardiovascular event; and 3) 1447 participants who were randomly selected from the full HUNT 3 cohort[5]. All

individuals provided informed written consent and the study was approved by the Regional Committee for Medical and Health Research Ethics (REK South-East C 2019/17355). The current study complies with STROBE guidelines.

### Protein measurement
Protein measurements in each analysis set were performed using a multiplexed Slow Off-rate Modified Aptamer (SOMAmer) assay (SOMAScan v4) as previously described[31]. Briefly, blood plasma was collected using standard protocols at ARIC sites during visits 3 and 5. Samples were stored at −80 °C and were sent to SomaLogic for quantification. The relative concentration levels of plasma proteins were determined using the SOMAmer assay, which utilizes small pieces of single strand DNA with modified aptamer regions. These aptamers are designed to bind to proteins based on a particular sequence or three-dimensional structure. DNA detection technology is then used to identify and quantify the proteins. Additional details on the assay and its performance have been previously published[72]. Relative protein concentration was available for 5,284 aptamers in total.

For SomaLogic assay quality control, ARIC data sets were combined with data from healthy controls and then normalized. Samples were flagged if at least one of the four sample calibration factors was outside of the acceptance criteria range of 0.4-2.5. These factors include hybridization control normalization factor, normalization factor for a dilution factor of 0.005%, normalization factor for a dilution factor of 0.5% and normalization factor for a dilution factor of 20%. SOMAmers were flagged if at least one intraplate calibration factor was out of the acceptance criteria of 0.8-1.2. Additional quality control procedures were implemented by the ARIC study. SOMAmer measures were $\log_2$ transformed to correct for skewness in the data distribution. Blind duplicates were run for 4% of participants in each visit, corresponding to 422 of 11565 individuals at visit 3 and 185 of 5327 individuals at visit 5. The median inter-assay coefficient of variation (CV) for SOMAmers measured at each visit were calculated using Bland-Altman analysis. For visit 3, this value was 6.31% and for visit 5 it was 4.67%. Quality control outliers were excluded as described below, and median split sample reliability coefficients were then calculated for each visit. At visit 3 the median reliability coefficient was 0.85, while at visit 5 the median reliability coefficient was a bit higher at 0.94. Additionally, manual annotation was completed for six UniProt IDs and three protein names.

Of the 5284 SOMAmers available before quality control exclusion, aptamers were excluded from analysis if they had a CV > 50% (55 at visit 3, 93 at visit 5). Additionally, proteins with variance <0.01 on a log-scale were excluded (12 at visit 3, 12 at visit 5). SOMAmers that bound to Fc mouse (228), contaminants (15), or non-proteins (70) including hybridization control elution, non-human proteins, non-biotin, non-cleavable, or spuriomer molecules were also excluded. Samples deemed to be outliers for each SOMAmer, defined as values outside of 5 times the standard deviation of the $\log_2$ scaled same mean, were winsorized. After quality control, a total of 4877 aptamers measuring levels of 4697 unique proteins were present for analysis.

In HUNT 3, non-fasting plasma samples were collected in EDTA tubes at random times of day. The samples were centrifuged, plasma aspirated and frozen at −80 °C within 24 h of blood draw. Never-thawed samples were shipped to SomaLogic, Boulder, CO for proteomic profiling using the equivalent assay (SOMAScan v4). Nine samples failed SOMALogic quality control and were excluded from further analyses. Only the 4877 aptamers passing ARIC quality control assessments were analyzed.

### Validation of candidate HF-associated aptamers
To provide orthogonal validation of key candidate proteins identified in this analysis, we measured plasma proteomics in a subset of 113 participant plasma samples at Visit 5 using the Olink Explore 3072

platform (2926 unique proteins) which uses multiplexed proximity extension assays (PEA)[73]. The assay sensitivity is comparable to traditional enzyme-linked immunosorbent assays (ELISAs)[74]. Of the 37 key candidate proteins identified in this analysis, 28 were captured in the Olink platform. ELISA assay was performed for 1 additional protein not captured by the Olink assay (SVEP1, AFG Bioscience sandwich ELISA assay) and high sensitivity troponin T was previously measured in ARIC Visit 5 sample (Elecsys high-sensitivity assay on an automated Cobas e411 analyzer, Roche Diagnostics®)[75].

## Covariate assessment

In ARIC, sex and race were self-reported. Hypertension was classified based on self-reported medication use or blood pressure ≥140/90 mmHg at any ARIC visit. Diabetes was defined based on self-report of a physician diagnosis of diabetes, anti-diabetic medication use, fasting glucose ≥126 mg/dL, or non-fasting glucose ≥200 mg/dL, at any ARIC visit. Smoking status was assessed at each visit based on interviewer-administered questionnaire. Body mass index (BMI) was assessed based on height and weight measures at each visit. Estimated glomerular filtration rate (eGFR) was assessed using the CKD-EPI equation using plasma creatinine measured at each visit[76]. Since study inception, ARIC participants have undergone surveillance for cardiovascular events including incident coronary heart disease events (definite or probable MI, or coronary revascularization) as previously described[77,78]. Atrial fibrillation was ascertained based on ECGs at 5 study visits and hospital discharge records as previously described[79]. Prevalent HF was based on hospitalization ICD codes prior to 2005[78] with additional physician adjudication since 2005 as previously described[80].

In HUNT, sex, race, diabetes, and smoking status were self-reported. Prevalent HF was defined by self-reported HF any time in life before the HUNT 3 visit. Prevalent CAD was defined by self-reported myocardial infarction, angina, percutaneous coronary intervention, or coronary artery bypass grafting. Prevalent hypertension was defined by self-reported anti-hypertensive medication use or blood pressure ≥140 mmHg systolic or ≥90 mmHg diastolic at the HUNT 3 visit. Prevalent AF was defined by the use of ICD 10 code I48.* or ICD 9 code 427.3 at any time in hospital records. BMI was calculated from weight and height measured at the HUNT 3 visit.

## Ascertainment of incident heart failure

In ARIC, incident HF events following the mid-life baseline assessment (Visit 3) were based on active surveillance of HF-related hospitalizations or death based on annual participant calls, review of local hospital discharges for cohort participants, and health department death certifications. Incident HF event was defined as the first occurrence of a hospitalization with a HF ICD9 code 428 or ICD10 code I50 or death certificate with one of these codes[78]. Incident HF events through 10 year follow-up post-mid-life baseline were included in this analysis.

Incident HF events following the late-life baseline assessment (Visit 5) was based on ARIC HF Event Classification as previously described[80]. HF event ascertainment and adjudication was triggered by any HF-related ICD discharge code in any position. Following comprehensive abstraction of medical records, which included information on LVEF, two independent reviewers classified each case using ARIC classification guidelines. Consistent with prior epidemiologic studies[81], HFpEF was defined as adjudicated HF with LVEF ≥ 50% at the incident HF hospitalization and HFrEF if LVEF was <50% at incident HF hospitalization. Incident HF events were ascertained through December 2018. Incident HF events following late-life baseline did not overlap with events following mid-life baseline. In HUNT, incident HF from the HUNT 3 visit through 2018 was first defined by the same ICD codes as in ARIC, and then adjudicated by a blinded cardiologist with access to all hospital records according to the 2012 European Society of Cardiology Guidelines for HF diagnosis.

## Physiologic testing and measures in the ARIC late-life analysis set

Quantitative measurements of cardiac structure and function were performed by echocardiography in ARIC at the time of late-life baseline (Visit 5), the design and methods of which have previously been described including reproducibility metrics[82]. All studies were performed by a limited set of certified sonographers using a study-specific acquisition protocol and all qualitative measures were performed by trained analysts at the ARIC Echocardiography Reading Center (Boston, MA). Arterial stiffness was assessed by pulse wave velocity (PWV) using the automated waveform analyzer VP-1000 Plus (Omron, Kyoto, Japan) after participants were supine for 5–10 min[83]. Repeatability of these measures have been previously reported[84]. Lung function was assessed based on the following spirometric variables: $FEV_1$, FVC and their ratio as previously described[24]. Participants underwent bioelectric impedance (measured using the Tanita Body Composition Analyzer, TBF-300A) and percent body fat, fat mass and lean body mass were calculated[85]. Grip strength, a measure of upper limb function, was assessed as the maximum handgrip isometric effort from two attempts using a handheld dynamometer[22].

## Statistical approach

**Associations of proteins with incident HF, HF risk factors, and measures of cardiovascular and non-cardiovascular function.** To assess the association of individual proteins with incident HF, time to event analyses were performed using Cox proportional hazards models. Each protein was tested in a separate multivariable model that additionally adjusted for age, BMI, eGFR by CKD-EPI, race, sex, current smoking status, prevalent CAD, prevalent DM, prevalent AF, and prevalent hypertension. Multiple testing correction was conducted using both the Bonferroni and FDR methods. Prior to testing, data from each analysis set were filtered to remove individuals with a negative or zero follow-up time and individuals with prevalent HF. For covariates with missing data, multiple imputation using chained equations was utilized to compute probable values. All imputed variables had less than 10% missingness. Protein levels were centered and scaled to a mean of 0 and a standard deviation of 1 to allow for comparison between the various protein models.

As a complementary feature selection approach, random forest models were constructed for ARIC visits 3 and 5 and for HUNT using the "randomForestSRC" R package (version 3.1.1)[86–88]. Outcome was defined as time to incident heart failure. Clinical covariates used in tree construction include age, BMI, eGFR by CKD-EPI, race, sex, current smoking status, prevalent CAD, prevalent DM, prevalent AF, and prevalent hypertension, as well as all normalized, scaled protein levels. First, trees were tuned to determine the number of variables to split at each node, and the minimum size of a terminal node that optimized the out-of-bag (OOB) error. Then these optimal values were utilized to construct a single random forest model. After constructing this optimal model, any proteins that did not reach the 80th percentile for depth were removed from the potential covariates list and the tree fitting process was repeated. 30 iterations were run in total and the tree with the smallest OOB error was chosen as the final random forest model for each of the three data sets.

As a complementary approach to the main parallel analyses, we conducted meta-analysis using the ARIC visit 3, ARIC visit 5, and HUNT data. Due to the correlated nature of ARIC visit 3 and visit 5 data, we first calculated a pooled ARIC HR estimate for association of protein level and time to heart failure using the Wei, Lin, and Weissfeld model[89]. Data was clustered by subject id and was stratified by ARIC visit (3 or 5). To fit the most "assumption-free" model, we allowed interactions between all covariates with the strata term, except for the protein level covariate. We corrected for the same set of covariates as in the original parallel analysis models. Meta-analysis was then conducted on the pooled ARIC HR and the HUNT HR using the inverse

variance method. Proteins that met a Bonferroni level of significance were used as input to overrepresentation pathway analysis using Qiagen IPA software.

The associations of individual proteins with incident HF risk factors were assessed using Cox proportional hazards models using the protein levels at ARIC Visit 3 were used, with outcomes censored at a subject's ARIC Visit 5 date. Individuals with prevalent risk factor diagnoses were excluded from analyses. Incident risk factor development was ascertained for each risk factor separately. Criteria for incident hypertension included measured BP > 140/90 mmHg, self-reported use of hypertension medications in the past 2 weeks, or if a patient indicated that a doctor has said they had high blood pressure since last contact during an annual follow-up call (AFU)[90]. Criteria for incident diabetes included fasting glucose >126 or participant report that a doctor has said they had diabetes or sugar in the blood since last contact during an AFU[91]. Incident coronary artery disease was defined as an adjudicated myocardial infarction or fatal CHD event after the Visit 3 date[77]. Criteria for incident CKD included an eGFR <60 mL/min/1.73 m^2 at ARIC Visits 4 or 5 accompanied by a ≥ 25% decline relative to the Visit 3 value, or kidney disease related hospitalization or death based on ICD9 codes 581-583, 585-589 and ICD10 codes N03, N04, N19, N25-N27[92]. Each protein was tested in a separate multivariate model that adjusted for age, BMI, eGFR by CKD-EPI, race, sex, current smoking status, prevalent CAD, prevalent DM, prevalent AF, and prevalent hypertension, excluding the adjustment for related prevalent risk factor. Multiple testing correction was conducted using FDR adjustment methods.

The cross-sectional associations of HF-related proteins with measures of cardiovascular and non-cardiovascular function were assessed at ARIC Visit 5 using multivariable linear regression. Continuous outcome measures were scaled to mean 0 and centered to a standard deviation of 1 prior to analysis. Regression models were adjusted for the covariates of age, BMI, eGFR by CKD-EPI, race, sex, current smoking status, prevalent CAD, prevalent DM, prevalent AF, and prevalent hypertension. For echocardiographic measures of cardiac structure and function, models were additionally adjusted for heart rate and systolic blood pressure at the time of echocardiography. Missing covariate data was imputed using multiple imputation using chained equations.

All analyses were conducted using R (v4.2.0).

**Mendelian randomization and colocalization analysis.** For proteins that were significantly associated with incident HF, we applied a two-sample Mendelian randomization (MR) approach[93,94] to assess the potential causal relationships between those proteins and their corresponding outcomes. To minimize bias, we used genetic summary statistics from two independent European samples for exposure and outcome data in each analysis. Instrumental variables (IVs) for protein quantitative trait loci (pQTLs) were obtained from three published studies: the INTERVAL study ($n = 3301$)[14], which included summary statistics for 2994 proteins measured by SOMAscan, the AGES cohort study ($n = 5368$)[15], with summary statistics for 4782 SOMAscan measured proteins, and Fenland study individuals with European descent ($n = 10,708$)[16], including summary statistics for 4775 SOMAscan protein targets. The summary statistics for HF were obtained from the HERMES consortium ($n = 977,323$)[95]. Summary statistics for cardiac structure/function outcomes (LVEDV, LVESV and LVEF) were obtained from UK Biobank ($n = 36,041$)[96]. IVs were selected if they reached genome-wide significance ($p < 5 \times 10^{-8}$), and the selected IVs were further clumped with r2 < 0.001. Wald tests or inverse variance weighted (IVW) tests, for more than one IV, were performed to calculate a causal estimate. If more than two IVs were included, Cochran's heterogeneity test was used to test the heterogeneity. If three or more IVs were included, the MR Egger method was utilized to test for the presence of horizontal pleiotropy[97]. 28 out of 36 HF-associated

proteins had IVs selected and 30 out of 36 echo- and/or HF-associated proteins had IVs selected respectively. The significance threshold for MR association was determined using Bonferroni correction for the number of independent sets of IVs tested, $p < 7e-4$ for HERMES HF (0.05/70), and $p < 5e-4$ for cardiac structure and function (0.05/92). For the observed significant associations, we further conducted a backward MR, which used IVs for cardiac structure and function as exposure and IVs for proteins as outcome, to detect potential reverse causality. All the analyses were performed using the R package "Two-SampleMR" (version 0.5.6)[98].

Two-sample MR was also utilized to test for potential causal relationships between the proteins found to be significantly associated with one of the tested HF risk factors, including AF, CHD, CKD, type II diabetes (DM), and hypertension. IVs for proteins were obtained using the same studies as previously described. The summary statistics for AF were obtained from a GWAS meta-analysis of 6 studies ($n = 1,030,836$)[99], summary statistics for CHD from UK Biobank and replicated using CARDIoGRAMplusC4D data ($n = 296,525$)[100], those for CKD from a 43 study GWAS meta-analysis ($n = 117,165$)[101], those for DM from a GWAS meta-analysis of 3 studies ($n = 655,666$)[102], and summary statistics for hypertension were from a UK Biobank GWAS stored in the IEU Open GWAS Project Database ($n = 463,010$)[103]. The same thresholds and testing methods were utilized as for the HF and cardiac structure/function analyses. Once again, multiple testing correction was utilized, resulting in significance thresholds of $p < 4.5e-4$ for AF and hypertension (0.05/112), $p < 4.7e-4$ for CHD (0.05/106), $p < 8.8e-4$ for CKD (0.05/57), and $p < 6.9e-4$ for DM (0.05/72). Backward MR was also conducted for any observed significant associations to look for potential reverse causality.

Sensitivity analyses were conducted for all combinations of associated proteins and their outcome. In one set of analyses, protein IVs were selected using a lower threshold ($p < 5 \times 10^{-4}$) and results were compared with the stricter threshold analyses in an attempt to assess the validity of the instrumental variable assumptions[104]. Analyses using only cis-acting pQTLs were also conducted (Supplementary Fig. 6).

For significant MR hits in a single genomic region (cis-MR) we performed colocalization analysis using the R package coloc (v5.1.0)[105]. Colocalization assuming a single causal variant and conditional colocalization using SuSiE were utilized[106,107]. For each cis-MR hit, we subset the full GWAS summary statistics to within 500 kb of the SNP of interest. We used the default priors for the coloc.abf() function. Evidence for colocalization was based on the posterior probability of H4 from software output, using a threshold of 75% to denote sufficient evidence for colocalization. For analyses conducted with coloc.susie, the credible set with the largest posterior probability of H4 was used. A H3 > 75% threshold was used to conclude that the protein and HF phenotype of interest are likely driven by different causal variants, which does not support colocalization.

**Consensus clustering analysis.** Consensus clustering was performed separately for ARIC visit 5 and HUNT datasets using the "ConsensusClusterPlus" R package (version 1.61.0)[108]. Scaled log-transformed protein levels for 10 proteins that had significant associations from the MR analyses were used as input for the clustering. We utilized the K-means algorithm for clustering and the Euclidean distance as a measure of distance between the subjects' protein levels. The number of clusters evaluated ranged from 2 to 5. For each of 200 subsamples, 80% of subject data was sampled and all protein data was sampled. The number of clusters present in the data was determined using diagnostic plots. To determine an optimal number of clusters, we look for clean separation in consensus matrix heatmaps, cluster consensus scores over 0.8 for all clusters and a low proportion of ambiguously clustered pairs. After determining the optimal number of clusters, differences in cluster features were examined by calculating the standardized mean difference in protein levels for each of the clusters. We

plotted Kaplan Meier survival curves and employed multivariable Cox proportional hazards models to examine whether cluster membership impacted incident HF risk. Models adjusted for age, gender, Field Center, hypertension, diabetes, BMI, atrial fibrillation, smoking status, CHD, and eGFR.

**Weighted correlation network analysis.** Weighted correlation network analysis was performed separately for ARIC visits 3 and 5 and HUNT datasets using the "WGCNA" R package (version 1.71)[18,109]. ARIC visit 5 data was first used to determine a soft threshold for the unsigned protein co-expression values. This threshold was utilized to transform the co-expression data into an adjacency matrix. Hierarchical clustering was then completed using the topological overlap matrix dissimilarity and modules were identified using dynamic branch cutting methods. Eigengenes were calculated for each module, separately for each of the three datasets. These eigengene values were then scaled to a mean of 0 and a standard deviation of 1. Next, we fit Cox proportional hazards models separately for ARIC visit 3 and visit 5 to determine if each of the module eigengenes were associated with time to incident HF.

**Genetic association with heart failure module proteins.** We conducted a genome-wide scan on the identified Brown Module in European and African Americans at visit 3. In ARIC, genotype was measured using Affymetrix Array 6.0 followed by imputation using the Trans-Omics for Precision Medicine (TOPMed) reference panel (freeze 5b)[110]. The detailed genotype quality control steps were described previously[111]. In brief, samples were excluded if they were first-degree relative of an included individual, outlier based on allele sharing and principal components analyses, or had sex mismatch. Variants were excluded if they had insufficient call rate (>5%), deviation from Hardy-Weinberg equilibrium ($p < 1e-5$), imputation quality <0.3 or minor allele frequency (MAF) < 0.5%. Linear regression was performed on the inverse normal transformed protein module eigenvector adjusting for age, sex, center and the first three principal components of ancestry to account for population stratification. Genome-wide significance was defined as $p < 5e-8$.

Many of the adjacent variants associated with the module eigenprotein may simply represent a single signal due to LD. To determine independent loci significantly associated with Brown Module, we first ascertained a 1-Mb region around each significant variant, and then, those 1-Mb regions with any overlap regions were merged. Two regions were identified to be associated with Brown Module: a 235 kb region on chromosome 1, which included 50 significant SNPs, and a 405 kb region on chromosome 2, which included 8 significant SNPs. We next used GCTA v1.93.2 to perform a stepwise model selection procedure to select independent variants in each region. A single COJO analysis using the "cojo-slct" option was carried out for each module. We also performed a LD-based clumping procedure using PLINK to the set of variants in a region. The COJO analysis and the LD-based clumping yielded two representative SNPs for the chromosome 1 region, rs10922100 and rs1061170, and one SNP for chromosome 2 region 2, rs1260326.

**Gene expression analysis.** Genotype-tissue expression data from the GTex project (Analysis Release V8 [dbGaP Accession phs000424.v8.p2]) and the Human Protein Atlas (HPA, proteinatlas.org) were used for analysis of genes coding for heart failure-related proteins, and their expression and localization across cardiac, vascular and metabolic tissue[12,13]. The expression of genes coding for heart failure-related proteins was quantified in heart, vascular, kidney, adipose, lung, and liver tissue as transcripts per million. Genes coding for heart failure-related proteins were organized using hierarchical clustering analysis. pQTLs with significant MR signal were tested using GTEx to determine whether they were also associated with gene expression levels in the analyzed

tissues. Significance was determined based on a given P-value threshold that is calculated separately for each differentially expressed gene in a tissue and which accounts for multiple testing using the false discovery rate method.

**Ingenuity pathway analysis.** Ingenuity Pathway Analysis (IPA) was performed to characterize the biological functions represented by the set of heart failure-associated proteins (QIAGEN IPA, QIAGEN, Hilden, Germany) and proteins within heart failure-associated network modules derived from weighted gene co-expression analysis. IPA uses the QIAGEN Knowledge Base, a database of manually curated scientific and medical content. Proteins-associated with incident heart failure in a meta-analysis of ARIC Visit 3, Visit 5 and HUNT3 at a Bonferroni-corrected two-sided P-value < 0.05 were subjected to IPA if the gene coding for the protein was available in the QIAGEN Knowledge Base. The beta-coefficient from Cox proportional hazards regression models and FDR-corrected P-values at ARIC Visit 3 were uploaded for each heart failure-associated protein. Certain proteins were targeted by multiple SOMAmers or multiple SOMAmers targeted different products of the same gene. The SOMAmer with the smallest beta-coefficient standard error was retained in IPA. IPA was conducted using all experimentally confirmed content for all species in the QIAGEN Knowledge Base, including proteins not measured by the SOMAScan platform. The reference dataset for p-value calculations was restricted to proteins measured by SOMAScan, however, to obtain the most accurate statistics on overrepresentation. Direct and indirect relationships were considered. Network analysis was restricted to a maximum of 35 molecules per network and 25 networks per analysis as recommended by QIAGEN and used previously in ARIC proteomics analysis[31]. Canonical pathways, networks and upstream regulators across the entire QIAGEN Knowledge Base were tested for overrepresentation within the set of heart failure-associated proteins relative to that expected with chance using a one-sided, right tailed Fisher's exact test (statistical significance was set at FDR-corrected $P < 0.05$)[112]. The match between expected and observed up- and down-regulation patterns is quantified using a Z-score[112].

**Identification of potential drug targets.** Candidate proteins with MR findings supporting a causal association with HF, HF risk factors, or cardiac structure/function were assessed as potential drug targets using the ChEMBL database and the druggable genome[17,55]. Both the database and the supplementary data of druggable genes were queried based on the HUGO gene name of the protein of interest.

**Reporting summary**
Further information on research design is available in the Nature Portfolio Reporting Summary linked to this article.

# Data availability

ARIC data access procedures are in accordance with participant informed consent and NIH data sharing policy. Anonymized data from the ARIC study are available at the NHLBI Biologic Specimen and Data Repository Information Coordinating Center and can be accessed through the website (https://biolincc.nhlbi.nih.gov/studies/aric/). Requests for access of ARIC data may also be submitted to the ARIC Publications Committee according to established study procedures which includes submission of a completed ARIC Manuscript Proposal From (available at https://aric.cscc.unc.edu/aric9/publications/policies_forms_and_guidelines) to the ARIC Publications Committee at aricpub@unc.edu. Review and approval of data access requests typically takes approximately one month. Sharing of ARIC data typically requires execution of a Data Use Agreement with the ARIC Coordinating Center at the University of North Carolina. The processed data are available in the Source Data file provided with this paper. The data generated in this study are provided in the

Supplementary Information/Source Data file. HUNT data are available on request. To protect participants' privacy, HUNT Research Centre aims to limit storage of data outside HUNT databank and cannot deposit data in open repositories. HUNT databank has precise information on all data exported to different projects and are able to reproduce these on request. There are no restrictions regarding data export given approval of applications to HUNT Research Centre. Further details are available at http://www.ntnu.edu/hunt/data. Source data are provided with this paper.

## Code availability
The code and statistical packages used for analyses in this study is available from the corresponding author upon request.

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

## Acknowledgements

The authors thank the staff and participants of the ARIC study and HUNT study for their important contributions. The Trøndelag Health Study (HUNT) is a collaboration between HUNT Research Centre (Faculty of Medicine and Health Sciences, Norwegian University of Science and Technology NTNU), Trøndelag County Council, Central Norway Regional Health Authority, and the Norwegian Institute of Public Health.

## Author contributions

A.M.S., P.L.M., T.O., and B.Y. designed the study. V.A., P.D., H.R., L.F.B., G.L., J.M., and N.Q.N. performed data analysis. R.C.H. and C.B. were responsible for laboratory assays. A.M.S., E.B., C.B., J.C., and B.Y. contributed data from ARIC. P.L.M., K.H., C.J., H.D., and T.O. contributed data from HUNT. A.M.S. and V.A. drafted the manuscript. All authors reviewed the manuscript and provided critical revision.

## Competing interests

The Atherosclerosis Risk in Communities study has been funded in whole or in part with Federal funds from the National Heart, Lung, and Blood Institute, National Institutes of Health, Department of Health and Human Services, under Contract nos. (HHSN268201700001I, HHSN268201700002I, HHSN268201700003I, HHSN268201700005I, HHSN268201700004I). Dr Shah was supported by NIH/NHLBI grants R01HL135008, R01HL143224, R01HL150342, R01HL148218, R01HL160025, and K24HL152008. Drs Myhre and Omland were supported by Research Council of Norway grant 296357/TOE. Dr Buckley was supported by NIH/NHLBI grant K23HL150311. Dr. Ballantyne was supported by NIH/NHLBI grant R01HL134320. Dr. Yu was in part supported by R01HL148218, R01 HL160793 and the JLH Foundation. The funder had no role design and conduct of this study; collection, management, analysis, and interpretation of the data; preparation, review, or approval of the manuscript; and decision to submit the manuscript for publication. Dr. Shah reports research support from Novartis through Brigham and Women's Hospital and consulting fees from Philips Ultrasound and Janssen. Dr. Myhre has consulted for and/or received speaker honoraria from Amarin, AmGen, Bayer, AstraZeneca, Boehringer-Ingelheim, Novartis, Novo Nordisk, Pharmacosmos, and Us2.ai. Dr Dalen

holds positions at Centre for Innovative Ultrasound Solutions (CIUS) and Precision Health Center for optimized cardiac care (ProCardio) - both Norwegian Research Council (NRC) centers for research-based innovation, where GE Healthcare, Horten, Norway is one of the institutional partners, has research contracts with GE Healthcare, and acts as research advisor for Boehringer Ingelheim. Dr. Hoogeveen reports research support from Denka Seiken through Baylor College of Medicine and consulting fees from Denka Seiken. Dr Ballantyne reports grant/research support from Abbott Diagnostic, Akcea, Amgen, Arrowhead, Esperion, Ionis, Merck, Novartis, Novo Nordisk, Regeneron, Roche Diagnostic, NIH, AHA, and ADA, and is a consultant for 89Bio, Abbott Diagnostics, Alnylam Pharmaceuticals, Althera, Amarin, Amgen, Arrowhead, Astra Zeneca, Denka Seiken, Esperion, Genentech, Gilead, Illumina, Ionis, Matinas BioPharma Inc, Merck, New Amsterdam, Novartis, Novo Nordisk, Pfizer, Regeneron, Roche Diagnostic. Dr. Coresh is a scientific advisor to Soma Logic. The remaining authors report no competing interests.
