## [Peer Review File · Nature Communications]

Large scale plasma proteomics identifies novel proteins and protein networks associated with heart failure developmentReviewers' Comments:

Reviewer #1:

Remarks to the Author:

In this study, Shah et al. perform a large-scale study investigating associations between 4877 proteins and heart failure in >13,900 heart failure (HF)-free individuals from the ARIC and HUNT studies. This is a well-written paper that will likely have a significant impact of the development and repurposing of medications to treat HF. I believe, however, that the study could benefit from some alterations in the study design to maximise the novelty and reliability of the findings.

Firstly, I wonder if a meta-analysis of observational associations between individual proteins and incident HF events would increase statistical power, likely resulting in more associations being discovered in the initial agnostic analysis of >4800 proteins. I understand that a meta-analysis can be performed also with random forest models (e.g. the 'metaforest' package in R). If the authors decide to keep separate analyses in the three datasets, I think that the Bonferroni threshold ($p < 0.05/N_{\text{tests}}$) should be $p < 0.05/(4877*3)$. The meta-analysis approach would enable the authors to retain the current Bonferroni threshold.

Secondly, I wonder if the authors have considered the possibility of performing colocalization analyses after cis-MR. When there are distinct but correlated causal variants for the protein and the outcome, cis-MR may provide a false positive result and this can be avoided with colocalization. A conditional colocalization approach such as PwCoCo or SuSiE would enable enhanced sensitivity by allowing more than one causal variant across the exposure and outcome datasets (e.g. see PMID: 35452592).

Minor comments

Figure 1 – the source datasets used for the MR could be mentioned in the figure or its legend. Also, at this stage it's not clear what the 3 analysis datasets are, as these are mentioned later and this Figure only shows two studies. Perhaps this information could also be mentioned in the legend.

It's not clear which protein-outcome associations are novel and which were found in previous studies (e.g. references #5-10 mentioned in the introduction). Perhaps you could add this information to the supplemental tables and further discuss the novelty of this study.

Row 628 – I believe you used Bonferroni correction as well.

Row 655 – I believe you meant "causal estimate"

In two-sample MR, you used HERMES as the source for genetic associations with HF, and UK Biobank (UKBB) for other outcomes related to cardiac structure and function. As some of the studies in HERMES/UKBB were also used to derive the genetic instruments, there is a possibility of winner's curse. Do the authors think that this may be an issue?

Reviewer #2:

Remarks to the Author:

Review of Shah et al.

Overview: This study describes the search for novel biomarkers of HF using a current state-of-the-art aptamer-based plasma proteomics platform to probe three large well-characterized clinical cohorts.

Strengths

The study offers several strengths. These include deep proteome characterization (5000 proteins) of plasma of across 11,000 patients from three cohorts. The manuscript centers on results that are reproducible across the cohorts. The study employs complementary analyses including a single-protein multivariate Cox proportional hazards model, a random survival forest model and weighted

gene network correlation again focusing proteins that emerge as biomarkers across methods. The analyses identified established biomarkers as well as novel ones. Mendelian Randomization (MR) is used to address the causal connection between candidate biomarkers, HF diagnoses, and risk factors. Reversed MR yielded no associations. The methods are described in considerable detail. Supplementary tables are informative.

Limitations

Some of the limitations are addressed in the discussion, including the potential for bias. Fears of target non-specificity are partially addressed by MR. Nevertheless, notwithstanding the strength of the Somascan platform (depth and high throughput), new protein biomarkers should be validated using an orthogonal method in a subset of samples (i.e., with validated antibodies, or preferably, targeted mass spectrometry (e.g. PRM)).

Minor issues

As presented, the resolution for most figures is inadequate

Overall, the size of the text is often smaller than optimal. This is particularly acute in Figure 7 where axes of graphs are unreadable.

Response to Reviewers

We thank the Editors and Reviewers for their insightful comments and suggestions. This letter details the changes to our manuscript that we have made in response to each of the comments and suggestions provided.

Reviewer #1

1. Firstly, I wonder if a meta-analysis of observational associations between individual proteins and incident HF events would increase statistical power, likely resulting in more associations being discovered in the initial agnostic analysis of >4800 proteins. I understand that a meta-analysis can be performed also with random forest models (e.g. the ‘metaforest’ package in R). If the authors decide to keep separate analyses in the three datasets, I think that the Bonferroni threshold ($p < 0.05/N_{\text{tests}}$) should be $p < 0.05/(4877*3)$. The meta-analysis approach would enable the authors to retain the current Bonferroni threshold.

Our objective in using a parallel analysis design in ARIC in mid-life (visit 3), ARIC late-life (visit 5), and HUNT was to identify proteins reproducibly associated with heart failure across heterogeneity due to age, geographic locations, and HF ascertainment, understanding that this approach would limit our power to detect associations. We do not agree that a Bonferroni threshold of $p < 0.05/(4877*3)$ would be appropriate for our analysis. By requiring a Bonferroni corrected $p < 0.05/4877$ (1.0×10^{-5}) in both ARIC and HUNT, the likelihood of a protein being significant in both would be $(1.0 \times 10^{-5}) * (1.0 \times 10^{-5}) = 1.0 \times 10^{-10}$, appreciably lower than 3.4×10^{-6} (equivalent to $0.05/[4877*3]$). Furthermore, we required protein associations with HF to be significant at 1.0×10^{-5} at two timepoints in ARIC, further reducing the likelihood of observing an association by chance (i.e. between $1.0 \times 10^{-10} [(1.0 \times 10^{-5})^2]$ and $1.0 \times 10^{-15} [(1.0 \times 10^{-5})^3]$ given the correlated nature of ARIC Visit 3 and Visit 5 data).

As suggested by the Reviewer, we have now performed meta-analysis using the ARIC visit 3, ARIC visit 5, and HUNT data. Due to the correlated nature of ARIC visit 3 and visit 5 data, we first calculated a pooled ARIC HR estimate for association of protein level and time to heart failure using the Wei, Lin, and Weissfeld model.¹ Data was clustered by subject id and was stratified by ARIC visit (3 or 5). To fit the most “assumption-free” model, we allowed interactions between all covariates with the strata term, except for the protein level covariate. We corrected for the same set of covariates as in the original parallel analysis models. Meta-analysis was then conducted on the pooled ARIC HR and the HUNT HR using the inverse variance method.

This meta-analysis identified 294 proteins significantly associated with incident heart failure at a Bonferroni-corrected significance level. As can be seen in the volcano plot, most of the candidate proteins identified in our parallel analysis represented the most strongly associated proteins from the meta-analysis. However, given the size of our ARIC sample relative to HUNT, the meta-analysis results did place greater weight on results from ARIC as opposed to HUNT. The ratio of weights (pooled ARIC weight/HUNT weight) ranged from 1 to 9.75 (mean 3), such

that the pooled ARIC estimate was given 3-fold the weight of the HUNT estimate on average. As ARIC differs from HUNT in sample age, race/ethnicity, and HF ascertainment, we therefore believe that our original – more conservative – parallel analysis identifies HF-associated proteins most likely to be generalizable.

We have now added the meta-analysis to the manuscript as a supplemental analysis. We also used the 294 proteins that met a Bonferroni level of significance as input to overrepresentation pathway analysis using Qiagen IPA software. We identified 3 pathways that meet FDR significance after multiple testing: (1) Hepatic Fibrosis / Hepatic Stellate Cell Activation; (2) LXR/RXR Activation; (3) Inhibition of Matrix Metalloproteases. The later 2 pathways were also enriched in our original analysis using the 150 proteins associated with incident HF in all three analysis sets at a nominal $p < 0.05$ as input to overrepresentation pathway analysis.

We have made the following changes to the manuscript:

Results section, 'Association of Protein Levels with Incident Heart Failure' sub-section, page 9:
Deleted: ~~“One hundred and fifty proteins were associated with incident HF in all three analysis sets at a nominal $p < 0.05$ and enriched for canonical pathways related to lipid metabolism (LXR/RXR activation) and extracellular matrix turnover (inhibition of matrix metalloproteinases) at an FDR $p < 0.05$ (Supplemental Table 1).”~~

Added: “As a complementary approach to the main parallel analyses, we conducted meta-analysis using the ARIC visit 3, ARIC visit 5, and HUNT data (see Methods) which identified 294 proteins associated with incident HF at a Bonferroni-corrected level of significance (Supplemental Figure X). Most of the candidate proteins identified in our parallel analysis were the most strongly associated with incident HF in the meta-analysis. Overrepresentation pathway analysis using these 294 proteins as input identified 3 overrepresented pathways at $FDR < 0.05$: Hepatic Fibrosis / Hepatic Stellate Cell Activation, LXR/RXR Activation, and Inhibition of Matrix Metalloproteases (Supplemental Table 1).”

Methods section, 'Statistical Approach' sub-section, 'Associations of proteins with incident HF, HF risk factors, and measures of cardiovascular and non-cardiovascular function' topic, paragraph 2, page 40:

“As a complementary approach to the main parallel analyses, we conducted meta-analysis using the ARIC visit 3, ARIC visit 5, and HUNT data. Due to the correlated nature of ARIC visit 3 and visit 5 data, we first calculated a pooled ARIC HR estimate for association of protein level and time to heart failure using the Wei, Lin, and Weissfeld model.⁸⁹ Data was clustered by subject id and was stratified by ARIC visit (3 or 5). To fit the most “assumption-free” model, we allowed interactions between all covariates with the strata term, except for the protein level covariate. We corrected for the same set of covariates as in the original parallel analysis models. Meta-analysis was then conducted on the pooled ARIC HR and the HUNT HR using the inverse variance method. Proteins that met a Bonferroni level of significance were used as input to overrepresentation pathway analysis using Qiagen IPA software.”

Methods section, ‘Statistical Approach’ sub-section, ‘Ingenuity Pathway Analysis’ topic, page 48: “Proteins-associated with incident heart failure after in a meta-analysis of ARIC Visit 3, Visit 5 and in HUNT3 at an FDR a Bonferroni-corrected two-sided P-value < 0.05 were subjected to IPA if the gene coding for the protein was available in the QIAGEN Knowledge Base.”

Supplemental Figure 4: Volcano plot of the association of individual plasma proteins with incident HF based on meta-analysis of ARIC visit 3, ARIC visit 5, and HUNT data. Y-axis lines indicate significance threshold for FDR <0.05 (lower line) and Bonferroni significance (upper line). Red data points represent proteins identified in the primary parallel analysis.

Supplemental Table 1 (updated): Top enriched pathways from IPA analysis for the set of proteins associated with HF at Bonferroni significance based on meta-analysis of ARIC visit 3, ARIC visit 5, and HUNT data.

Ingenuity Canonical Pathways	-log(B-H p-value)	B-H p-value	Ratio	Molecules
Hepatic Fibrosis / Hepatic Stellate Cell Activation	1.37	0.04265795	0.121	BAMBI, COL13A1, CSF1, CXCL9, EGFR, FAS, IL6, KDR, MMP1, MYL6B, TIMP2, TNFRSF11B, VEGFA, VEGFD
LXR/RXR Activation	1.37	0.04265795	0.133	AHSG, APOA2, APOF, IL6, ITIH4, MSR1, PLTP, SAA1, SAA2, SERPINF2, TNFRSF11B
Inhibition of Matrix Metalloproteases	1.37	0.04265795	0.231	MMP1, MMP12, MMP19, THBS2, TIMP2, TIMP4

2. Secondly, I wonder if the authors have considered the possibility of performing colocalization analyses after cis-MR. When there are distinct but correlated causal variants for the protein and the outcome, cis-MR may provide a false positive result and this can be avoided with colocalization. A conditional colocalization approach such as PwCoCo or SuSiE would enable enhanced sensitivity by allowing more than one causal variant across the exposure and outcome datasets (e.g. see PMID: 35452592).

As suggested by the Reviewer, we have now performed colocalization analysis for significant MR hits in a single genomic region (cis-MR) using the R package coloc (v5.1.0).² Colocalization assuming a single causal variant and conditional colocalization using SuSiE^{3,4} were utilized. For each cis-MR hit, we subset the full GWAS summary statistics to within 500 kb of the SNP of interest and used the default priors for the coloc.abf() function. Evidence for colocalization was based on the posterior probability of H4 from software output, using a threshold of 75% to denote sufficient evidence for colocalization. For analyses conducted with coloc.susie, the credible set with the largest posterior probability of H4 was used.

As summarized in the Table below, the majority of MR significant associations had evidence for colocalization.

Protein	Outcome							
	HF	HTN	AF	DM	CHD	LVEDV	LVESV	LVEF
SVEP1	H4		H3					
SPON1		H4	H4			H4	H4	H4
NPPB		H3						
CCL15		H4			H4	H4	H4	
ITIH3		H3				H3		
FSTL3				H4		H4		
NRP1				H4	H4	H1		
ANGPTL3						H1		
IGFBP7							H3	H3
MFAP4								H1

H4 – evidence for colocalization (posterior probability of colocalization >75%); H3 – findings do not support colocalization; H1 – low power to assess for colocalization.
 HF – heart failure; HTN – hypertension; AF – atrial fibrillation; DM – diabetes; CHD – coronary heart disease; LVEDV – left ventricular (LV) end-diastolic volume; LVESV – LV end-systolic volume; LVEF – LV ejection fraction

We have made the following changes to the manuscript:

Results section, ‘Mendelian Randomization Causally Implicates Proteins’ sub-section, page 18: Colocalization analysis demonstrated evidence of colocalization for the majority of MR significant associations (21 of 33; Supplemental Table 7). We observed evidence of colocalization for observed MR associations for SPON1, CCL15, and FSTL3, for SVEP1 with HF, and for NRP1 with diabetes and CHD. Power was inadequate for colocalization for ANGPTL3 and

MFAP4. Our findings were negative for colocalization, suggesting two different causal variants for protein level and outcome, for the observed MR associations for NPPB, ITIH3, IGFBP7, and for SVEP1 with atrial fibrillation.’

Methods section, ‘Statistical Approach’ sub-section, ‘Mendelian Randomization Analysis’ (now revised to ‘Mendelian Randomization and Colocalization Analysis’) topic, paragraph 4, page 45:

‘For significant MR hits in a single genomic region (cis-MR) we performed colocalization analysis using the R package coloc (v5.1.0).¹⁰⁵ Colocalization assuming a single causal variant and conditional colocalization using SuSiE were utilized.^{106,107} For each cis-MR hit, we subset the full GWAS summary statistics to within 500 kb of the SNP of interest. We used the default priors for the coloc.abf() function. Evidence for colocalization was based on the posterior probability of H4 from software output, using a threshold of 75% to denote sufficient evidence for colocalization. For analyses conducted with coloc.susie, the credible set with the largest posterior probability of H4 was used.’

Supplemental Table 7: Results of colocalization analysis for significant MR associations

Outcome	SeqId	Protein	Full Name	SNP	Chr	Region Start (bp)	Region End (bp)	Highest Posterior Probability	Meets Significance Threshold (75%)
HF	SeqId_11109_56	SVEP1	Sushi, von Willebrand factor type A, EGF and pentraxin domain-containing protein 1	rs6687621	9	135637065	136637065	H4 (80.8%)	Yes
Hypertension	SeqId_16751_15	NPPB	Natriuretic peptides B	rs198379	1	11415467	12415467	H3 (~100%)	Yes
Hypertension	SeqId_4297_62	SPON1	Spondin-1	rs601338	19	48706674	49706674	H4 (97.2%)	Yes
Hypertension	SeqId_18289_16	CCL15	C-C motif chemokine 15	rs681343	19	48706462	49706462	H4 (97.6%)	Yes
Hypertension	SeqId_7145_1	ITIH3	Inter-alpha-trypsin inhibitor heavy chain H3	rs233529	3	52333219	53333219	H3 (61.0%)	Yes
Hypertension	SeqId_16751_15	NPPB	Natriuretic peptides B	rs198375	1	11413757	12413757	H3 (~100%)	Yes
Hypertension	SeqId_18289_16	CCL15	C-C motif chemokine 15	rs492602	19	48706417	49706417	H4 (97.2%)	Yes
A Fib	SeqId_11109_56	SVEP1	Sushi, von Willebrand factor type A, EGF and pentraxin domain-containing protein 1	rs45444697	1	154534632	155534632	H3 (~100%)	Yes
A Fib	SeqId_4297_62	SPON1	Spondin-1	rs10832169	11	13566486	14566486	H4 (92.8%)	Yes
A Fib	SeqId_4297_62	SPON1	Spondin-1	rs1969539	11	13538621	14538621	H4 (90.7%)	Yes
Diabetes	SeqId_3438_10	FSTL3	Follistatin-related protein 3	rs1260326	2	27230940	28230940	H4 (99.8%)	Yes
Diabetes	SeqId_5542_22	NRP1	Neuropilin-1	rs4665972	2	27098097	28098097	H4 (99.5%)	Yes
Diabetes	SeqId_5542_22	NRP1	Neuropilin-1	rs1260326	2	27230940	28230940	H4 (99.1%)	Yes
CHD	SeqId_18289_16	CCL15	C-C motif chemokine 15	rs28929474	14	94344947	95344947	H4 (99.8%)	Yes
CHD	SeqId_7145_1	ITIH3	Inter-alpha-trypsin inhibitor heavy chain H3	rs233529	3	52333219	53333219	H3 (80.7%)	Yes
CHD	SeqId_5542_22	NRP1	Neuropilin-1	rs2506150	10	32983308	33983308	H4 (56.3%)	No
CHD	SeqId_5542_22	NRP1	Neuropilin-1	rs2506149	10	32980713	33980713	H4 (73.6%)	No
CHD	SeqId_5542_22	NRP1	Neuropilin-1	rs4846913	1	229794715	230794715	H4 (84.3%)	Yes
CHD	SeqId_5542_22	NRP1	Neuropilin-1	rs10864728	1	229804914	230804914	H4 (90.5%)	Yes
LVEDV	SeqId_3438_10	FSTL3	Follistatin-related protein 3	rs1260326	2	27230940	28230940	H4 (62.9%)	No
LVEDV	SeqId_10391_1	ANGPT1	Angiotensinogen-related protein 3	rs74076477	1	62266601	63266601	H1 (99.8%)	Yes
LVEDV	SeqId_18289_16	CCL15	C-C motif chemokine 15	rs28929474	14	94344947	95344947	H4 (99.6%)	No
LVEDV	SeqId_5542_22	NRP1	Neuropilin-1	rs4665972	2	27098097	28098097	H1 (58.4%)	No
LVEDV	SeqId_5542_22	NRP1	Neuropilin-1	rs1260326	2	27230940	28230940	H1 (50.3%)	No
LVEDV	SeqId_4297_62	SPON1	Spondin-1	rs10832169	11	13566486	14566486	H4 (99.8%, with SuSiE)	Yes
LVEDV	SeqId_4297_62	SPON1	Spondin-1	rs1969539	11	13538621	14538621	H1 (53.6%)	No
LVESV	SeqId_3320_49	IGFBP7	Insulin-like growth factor-binding protein 7	rs9271147	6	32077385	33077385	H3 (87.9%)	Yes
LVESV	SeqId_18289_16	CCL15	C-C motif chemokine 15	rs28929474	14	94344947	95344947	H4 (99.4%)	Yes
LVESV	SeqId_4297_62	SPON1	Spondin-1	rs10832169	11	13566486	14566486	H4 (97.3%)	Yes
LVESV	SeqId_4297_62	SPON1	Spondin-1	rs1969539	11	13538621	14538621	H4 (97.3%)	Yes
LVEF	SeqId_3320_49	IGFBP7	Insulin-like growth factor-binding protein 7	rs9271147	6	32077385	33077385	H3 (76.9%)	Yes
LVEF	SeqId_5636_10	MFAP4	Microfibril-associated glycoprotein 4	rs139356332	17	18789286	19789286	H1 (71.3%)	No
LVEF	SeqId_4297_62	SPON1	Spondin-1	rs10832169	11	13566486	14566486	H4 (95.0%)	Yes
LVEF	SeqId_4297_62	SPON1	Spondin-1	rs1969539	11	13538621	14538621	H4 (94.9%)	Yes

3. Figure 1 – the source datasets used for the MR could be mentioned in the figure or its legend. Also, at this stage it’s not clear what the 3 analysis datasets are, as these are mentioned later and this Figure only shows two studies. Perhaps this information could also be mentioned in the legend.

We have now updated the legend to Figure 1 to specify the analysis sets being used and the datasets used for MR analysis, as follows:

Figure 1, legend, page 6:

“**Figure 1.** Schematic overview of study design. The three analysis sets are: ARIC Visit 3 (1993-1995; age 60±5 years, 54% women, 21% Black race), ARIC Visit 5 (2011-2013; age 75±5 years,

58% women, 17% Black race), and HUNT cycle 3 (2006-2008; age 65±10 years, 39% women, 0% Black race). For MR analysis, protein quantitative trait loci (pQTLs) were obtained from the INTERVAL, AGES, and Fenland studies as instrument variables (IV). Summary statistics for heart failure were from the HERMES study and summary statistics for cardiac structure and function were from UK Biobank.”

4. It’s not clear which protein-outcome associations are novel and which were found in previous studies (e.g. references #5-10 mentioned in the introduction). Perhaps you could add this information to the supplemental tables and further discuss the novelty of this study.

While pre-clinical data supports the involvement of most of the candidate proteins in cardiac dysfunction or HF development, only a minority have previously been identified as circulating biomarkers of risk for incident HF. Excluding NT-proBNP and troponin T, circulating levels of 8 of the 35 proteins identified in this analysis have previously been associated with risk of incident HF. Notably, circulating levels of 10 of these 35 proteins have previously been associated with risk of adverse outcomes among patients with prevalent HF (HFpEF, HFrEF, or both). As suggested by the Reviewer, we have now added this information as Supplemental Table X to the manuscript. We have also added the following to the manuscript text:

Discussion section, paragraph 1, page 27:

“While pre-clinical data supports the involvement of most of these in HF development, circulating levels of only 8 have previously been associated with risk of incident HF and 10 with risk of adverse outcomes among patients with prevalent HF (excluding NT-proBNP and troponin T; Supplemental Table 12).”

Supplemental Table 12. Previously described associations of candidate proteins with incident HF and clinical outcomes among patients with prevalent HF (HF overall, HFpEF, HFrEF).

Candidate Protein	Incident HF	Outcomes in Prevalent HF
SVEP1		HFrEF ⁵
SPON1	HF ⁶⁻⁸	HFrEF ⁹ , HFpEF ¹⁰
FSTL3		
APOF		
ANGPTL3		
IGFBP7		HFrEF ⁹ , HFpEF ¹¹
MFAP4		
CCL15		
ITIH3		
NRP1		
WFDC2		
ANGPT2		HFpEF ¹² , HFrEF ⁵ , HF overall ¹³
GDF15	HF ⁶⁻⁸	HFrEF ^{5,9} , HFpEF ¹⁰⁻¹²
THBS2	HF ^{14,15}	HFrEF ^{5,16} , HF ¹³
CILP2		
GHR	HF ¹⁷	

TAGLN		
IGFBP2	HF ^{15, 17}	
TREM1		
FSTL1		HFrEF ¹⁸
TIMP4		
C9	HF ¹⁴	
SLIT2		
TNFRSF11B	HF ^{6, 8}	HFpEF ^{10, 12}
CELA1		
EGFR		HFrEF ^{9, 19}
CACNA2D3		
CLEC3B (tetranectin)		
PTPRD		
ATP1B1		
MSR1		
FBLN5		
MMP12	HF ⁶⁻⁸	HFpEF ¹⁰
IGDCC4		
SLITRK1		

5. Row 628 – I believe you used Bonferroni correction as well.

For the association of candidate proteins with incident HF risk factors, we accounted for multiple testing using FDR correction (row 628). For association of proteins with incident HF, we conducted multiple testing correction using both Bonferroni and FDR methods (row 592).

6. Row 655 – I believe you meant “causal estimate”

We thank the Reviewer and have now corrected this typographical error.

7. In two-sample MR, you used HERMES as the source for genetic associations with HF, and UK Biobank (UKBB) for other outcomes related to cardiac structure and function. As some of the studies in HERMES/UKBB were also used to derive the genetic instruments, there is a possibility of winner’s curse. Do the authors think that this may be an issue?

We appreciate the Reviewer’s concern. In our analysis, our interpretation of the two-sample MR results focuses on the presence or absence of a statistically significant MR association, and not on the magnitude of that association. Therefore, while the winner’s curse may result in a larger effect size, we would not expect this to substantively impact the interpretation of our findings.

Reviewer #2

1. Some of the limitations are addressed in the discussion, including the potential for bias. Fears of target non-specificity are partially addressed by MR. Nevertheless, notwithstanding the strength of the Somascan platform (depth and high throughput), new protein biomarkers should be validated using an orthogonal method in a subset of samples (i.e., with validated antibodies, or preferably, targeted mass spectrometry (e.g. PRM)).

To provide orthogonal validation of the majority of key candidate proteins identified in this analysis, we have now additionally measured plasma proteomics in a subset of 113 participant plasma samples at Visit 5 using the Olink Explore 3072 platform which measures 2,926 unique proteins. The Olink platform uses multiplexed proximity extension assays (PEA), where two matched antibodies labelled with unique DNA oligonucleotides simultaneously bind to a target protein, allowing their DNA oligonucleotides to hybridize and serve as a template for a DNA polymerase-dependent extension step.²⁰ The assay sensitivity is comparable to traditional enzyme-linked immunosorbent assays (ELISAs).²¹ Of the 37 key candidate proteins identified in this analysis, 28 were captured in the Olink platform (Figure). ELISA assay was performed for 1 additional protein not captured by the Olink assay (SVEP1, AFG Bioscience sandwich ELISA assay) and high sensitivity troponin T was previously measured in ARIC Visit 5 sample (Elecsys high-sensitivity assay on an automated Cobas e411 analyzer, Roche Diagnostics®).²² No validation was available for 7/37 proteins: TAGLN, TREM1, CACNA203, FBLN5, CLIP2, CELA1, PTPRD. As shown in the Figure, correlation was good (>0.70) for the majority of proteins (18/30) assessed, moderate (0.40-0.70) for 8/30, and poor (<0.40) for only 4 (FSTL1, APOF, SVEP1, ATP1B1).

We have now added this Figure to the Supplemental Materials at Supplemental Figure 6. We have also made the following additional changes to the manuscript text:

Results section, 'Association of protein levels with incident heart failure' sub-section, paragraph 3, page 10:

"We used an orthogonal method to validate aptamer specificity (Olink Explore 3072 proximity extension assay [n=27], targeted ELISAs [n=2], or electrochemiluminescence sandwich immunoassay [n=1]) using plasma from a subset of 113 participants (see Supplemental Figure 6). No validation was available for 7/37 proteins: TAGLN, TREM1, CACNA203, FBLN5, CLIP2, CELA1, PTPRD. Correlation was good (>0.70) for the majority of proteins (18/30) assessed, moderate (0.40-0.70) for 8/30, and poor (<0.40) for only 4 (FSTL1, APOF, SVEP1, ATP1B1)."

Methods section, 'Validation of candidate HF-associated aptamers' sub-section, page 37:

"To provide orthogonal validation of key candidate proteins identified in this analysis, we measured plasma proteomics in a subset of 113 participant plasma samples at Visit 5 using the Olink Explore 3072 platform (2,926 unique proteins) which uses multiplexed proximity extension assays (PEA).²⁰ The assay sensitivity is comparable to traditional enzyme-linked immunosorbent assays (ELISAs).²¹ Of the 37 key candidate proteins identified in this analysis, 28 were captured in the Olink platform. ELISA assay was performed for 1 additional protein not captured by the

Olink assay (SVEP1, AFG Bioscience sandwich ELISA assay) and high sensitivity troponin T was previously measured in ARIC Visit 5 sample (Elecsys high-sensitivity assay on an automated Cobas e411 analyzer, Roche Diagnostics®).²²

Supplemental Figure 6: Scatter plots and associated Pearson correlations for Somascan aptamer levels (Y-axis) and orthogonal antibody-based assay (X-axis; see Methods). Values from all assays were log2-transformed.

2. As presented, the resolution for most figures is inadequate

We have now updated the manuscript figures to ensure that image resolution is higher.

3. Overall, the size of the text is often smaller than optimal. This is particularly acute in Figure 7 where axes of graphs are unreadable.

We have now revised Figure 7 to increase the text to enable better readability.

References Cited

1. Wei LJ, Lin, D. Y., & Weissfeld, L. Regression analysis of multivariate incomplete failure time data by modeling marginal distributions. *Journal of the American Statistical Association*. 1989;84:1065-1073.
2. Giambartolomei C, Vukcevic D, Schadt EE, Franke L, Hingorani AD, Wallace C and Plagnol V. Bayesian test for colocalisation between pairs of genetic association studies using summary statistics. *PLoS Genet*. 2014;10:e1004383.
3. Wallace C. A more accurate method for colocalisation analysis allowing for multiple causal variants. *PLoS Genet*. 2021;17:e1009440.
4. Zou Y, Carbonetto P, Wang G and Stephens M. Fine-mapping from summary data with the "Sum of Single Effects" model. *PLoS Genet*. 2022;18:e1010299.
5. Zhang L, Cunningham JW, Claggett BL, Jacob J, Mendelson MM, Serrano-Fernandez P, Kaiser S, Yates DP, Healey M, Chen CW, Turner GM, Patel-Murray NL, Zhao F, Beste MT, Laramie JM, Abraham WT, Jhund PS, Kober L, Packer M, Rouleau J, Zile MR, Prescott MF, Lefkowitz M, McMurray JJV, Solomon SD and Chutkow W. Aptamer Proteomics for Biomarker Discovery in Heart Failure With Reduced Ejection Fraction. *Circulation*. 2022;146:1411-1414.
6. Stenemo M, Nowak C, Byberg L, Sundstrom J, Giedraitis V, Lind L, Ingelsson E, Fall T and Arnlov J. Circulating proteins as predictors of incident heart failure in the elderly. *Eur J Heart Fail*. 2018;20:55-62.
7. Lind L, Arnlov J and Sundstrom J. Plasma Protein Profile of Incident Myocardial Infarction, Ischemic Stroke, and Heart Failure in 2 Cohorts. *J Am Heart Assoc*. 2021;10:e017900.
8. Henry A, Gordillo-Maranon M, Finan C, Schmidt AF, Ferreira JP, Karra R, Sundstrom J, Lind L, Arnlov J, Zannad F, Malarstig A, Hingorani AD, Lumbers RT, Hermes and Consortia S. Therapeutic Targets for Heart Failure Identified Using Proteomics and Mendelian Randomization. *Circulation*. 2022;145:1205-1217.
9. Tromp J, Westenbrink BD, Ouwerkerk W, van Veldhuisen DJ, Samani NJ, Ponikowski P, Metra M, Anker SD, Cleland JG, Dickstein K, Filippatos G, van der Harst P, Lang CC, Ng LL, Zannad F, Zwinderman AH, Hillege HL, van der Meer P and Voors AA. Identifying Pathophysiological Mechanisms in Heart Failure With Reduced Versus Preserved Ejection Fraction. *J Am Coll Cardiol*. 2018;72:1081-1090.
10. Hage C, Michaelsson E, Linde C, Donal E, Daubert JC, Gan LM and Lund LH. Inflammatory Biomarkers Predict Heart Failure Severity and Prognosis in Patients With Heart Failure With Preserved Ejection Fraction: A Holistic Proteomic Approach. *Circ Cardiovasc Genet*. 2017;10.
11. Sanders-van Wijk S, Tromp J, Beussink-Nelson L, Hage C, Svedlund S, Saraste A, Swat SA, Sanchez C, Njoroge J, Tan RS, Fermer ML, Gan LM, Lund LH, Lam CSP and Shah SJ. Proteomic Evaluation of the Comorbidity-Inflammation Paradigm in Heart Failure With Preserved Ejection Fraction: Results From the PROMIS-HFpEF Study. *Circulation*. 2020;142:2029-2044.
12. Chirinos JA, Orlenko A, Zhao L, Basso MD, Cvijic ME, Li Z, Spires TE, Yarde M, Wang Z, Seiffert DA, Prenner S, Zamani P, Bhattacharya P, Kumar A, Margulies KB, Car BD, Gordon DA, Moore JH and Cappola TP. Multiple Plasma Biomarkers for Risk Stratification in Patients With Heart Failure and Preserved Ejection Fraction. *J Am Coll Cardiol*. 2020;75:1281-1295.
13. Wells QS, Gupta DK, Smith JG, Collins SP, Storrow AB, Ferguson J, Smith ML, Pulley JM, Collier S, Wang X, Roden DM, Gerszten RE and Wang TJ. Accelerating Biomarker Discovery

- Through Electronic Health Records, Automated Biobanking, and Proteomics. *J Am Coll Cardiol*. 2019;73:2195-2205.
14. Egerstedt A, Berntsson J, Smith ML, Gidlof O, Nilsson R, Benson M, Wells QS, Celik S, Lejonberg C, Farrell L, Sinha S, Shen D, Lundgren J, Radegran G, Ngo D, Engstrom G, Yang Q, Wang TJ, Gerszten RE and Smith JG. Profiling of the plasma proteome across different stages of human heart failure. *Nat Commun*. 2019;10:5830.
 15. Naylor M, Short MI, Rasheed H, Lin H, Jonasson C, Yang Q, Hveem K, Felix JF, Morrison AC, Wild PS, Morley MP, Cappola TP, Benson MD, Group CH-HFW, Consortium CH-E, Ngo D, Sinha S, Keyes MJ, Shen D, Wang TJ, Larson MG, Brumpton BM, Gerszten RE, Omland T and Vasani RS. Aptamer-Based Proteomic Platform Identifies Novel Protein Predictors of Incident Heart Failure and Echocardiographic Traits. *Circ Heart Fail*. 2020;13:e006749.
 16. Hanatani S, Izumiya Y, Takashio S, Kimura Y, Araki S, Rokutanda T, Tsujita K, Yamamoto E, Tanaka T, Yamamuro M, Kojima S, Tayama S, Kaikita K, Hokimoto S and Ogawa H. Circulating thrombospondin-2 reflects disease severity and predicts outcome of heart failure with reduced ejection fraction. *Circ J*. 2014;78:903-10.
 17. Katz DH, Tahir UA, Ngo D, Benson MD, Gao Y, Shi X, Naylor M, Keyes MJ, Larson MG, Hall ME, Correa A, Sinha S, Shen D, Herzig M, Yang Q, Robbins JM, Chen ZZ, Cruz DE, Peterson B, Vasani RS, Wang TJ, Wilson JG and Gerszten RE. Multiomic Profiling in Black and White Populations Reveals Novel Candidate Pathways in Left Ventricular Hypertrophy and Incident Heart Failure Specific to Black Adults. *Circ Genom Precis Med*. 2021;14:e003191.
 18. El-Armouche A, Ouchi N, Tanaka K, Doros G, Wittkopper K, Schulze T, Eschenhagen T, Walsh K and Sam F. Follistatin-like 1 in chronic systolic heart failure: a marker of left ventricular remodeling. *Circ Heart Fail*. 2011;4:621-7.
 19. Gui H, She R, Luzum J, Li J, Bryson TD, Pinto Y, Sabbah HN, Williams LK and Lanfear DE. Plasma Proteomic Profile Predicts Survival in Heart Failure With Reduced Ejection Fraction. *Circ Genom Precis Med*. 2021;14:e003140.
 20. Smith JG and Gerszten RE. Emerging Affinity-Based Proteomic Technologies for Large-Scale Plasma Profiling in Cardiovascular Disease. *Circulation*. 2017;135:1651-1664.
 21. Arrigo M, Vodovar N, Von Moos S, Masson E, Segerer S, Cippa PE and Mebazaa A. High accuracy of proximity extension assay technology for the quantification of plasma brain natriuretic peptide. *J Clin Lab Anal*. 2018;32:e22574.
 22. McEvoy JW, Chen Y, Ndumele CE, Solomon SD, Nambi V, Ballantyne CM, Blumenthal RS, Coresh J and Selvin E. 6-Year Change in High Sensitivity Cardiac Troponin-T and Risk for Subsequent Coronary Heart Disease, Heart Failure and Death. *JAMA cardiology*. 2016;1:519-528.

Reviewers' Comments:

Reviewer #1:

Remarks to the Author:

I appreciate the authors' attempts to address my concerns. I agree with most of their remarks and have only a couple of major outstanding points:

1) A $H4 > 75\%$ threshold would be better to conclude evidence of colocalization. $H3 > 75\%$ would indicate that the protein and HF phenotype of interest are likely driven by different causal variants, which does not provide evidence for colocalization (e.g. see PMID:35452592). Based on ST7 results, using $H4$ would still yield 12 colocalizing signals among 33 MR-significant associations.

2) While it is true that the winner's curse may result in larger effect size, a larger effect size is less likely to be obtained by chance, which implies a potential impact on statistical significance. I think that the authors should add this potential limitation to the Discussion section.

Reviewer #2:

Remarks to the Author:

The authors have addressed my primary concern that key results should be confirmed with an orthogonal method.

Response to Reviewers

We thank the Editors and Reviewers for their insightful comments and suggestions. This letter details the changes to our manuscript that we have made in response to each of the comments and suggestions provided.

Reviewer #1

1. A H4>75% threshold would be better to conclude evidence of colocalization. H3>75% would indicate that the protein and HF phenotype of interest are likely driven by different causal variants, which does not provide evidence for colocalization (e.g. see PMID:35452592). Based on ST7 results, using H4 would still yield 12 colocalizing signals among 33 MR-significant associations.

We agree with the Reviewer and did use a H4 >75% threshold to conclude evidence of colocalization, and a H3 >75% threshold to conclude that the protein and phenotype of interest are likely driven by different causal variants. As denoted in Supplemental Table 7, among the 34 MR-significant associations, 19 variant-phenotype associations had evidence of colocalization based on these thresholds while 6 demonstrated evidence that different causal variants relate to the protein and outcome. In total, 13 of 23 protein-outcome relationships identified by MR demonstrated evidence of colocalization, while 5 of 23 demonstrated evidence of different causal variants. We apologize for the typographical errors in our prior Response and Revised manuscript, which have now been corrected in the manuscript. In addition, we have modified Supplemental Table 7 to replace the column 'Meets Significance Threshold for Highest Posterior Probability (75%)' with the column 'Evidence of colocalization (H4 >75%)'. The 'Meets Significance Threshold for Highest Posterior Probability (75%)' column reported whether the 75% threshold was met for whichever highest posterior probability (H1 through H4) was identified. In contrast, the new 'Evidence of colocalization (H4 >75%)' column reports whether both (a) the highest posterior probability was H4, and (b) the threshold of >75% was met.

We have made the following changes to the manuscript:

Results section, 'Mendelian Randomization Causally Implicates Proteins' sub-section, page 18: Colocalization analysis demonstrated evidence of colocalization for the majority of MR significant associations (19 of 34; Supplemental Table 7). We observed evidence of colocalization for observed MR associations for SPON1, CCL15, and FSTL3 with multiple outcomes; for SVEP1 with HF; and for NRP1 with diabetes and CHD. Power was inadequate for colocalization for ANGPTL3 and MFAP4. For the observed MR associations for NPPB, ITIH3, IGFBP7 with multiple outcomes and for the association of SVEP1 with atrial fibrillation, our findings were negative for colocalization, suggesting two different causal variants for protein level and outcome.'

Methods section, ‘Statistical Approach’ sub-section, ‘Mendelian Randomization and Colocalization Analysis’ topic, paragraph 4, page 45:

‘For significant MR hits in a single genomic region (cis-MR) we performed colocalization analysis using the R package coloc (v5.1.0).¹⁰⁵ Colocalization assuming a single causal variant and conditional colocalization using SuSiE were utilized.^{106,107} For each cis-MR hit, we subset the full GWAS summary statistics to within 500 kb of the SNP of interest. We used the default priors for the coloc.abf() function. Evidence for colocalization was based on the posterior probability of H4 from software output, using a threshold of 75% to denote sufficient evidence for colocalization. For analyses conducted with coloc.susie, the credible set with the largest posterior probability of H4 was used. **A H3 >75% threshold was used to conclude that the protein and HF phenotype of interest are likely driven by different causal variants, which does not support colocalization.**’

Supplemental Table 7: Results of colocalization analysis for significant MR associations

Outcome	SeqId	Protein	Full Name	SNP	Chr	Region Start (bp)	Region End (bp)	Highest Posterior Probability	Evidence of colocalization (H4 >75%)
HF	SeqId_11109_56	SVEP1	Sushi, von Willebrand factor type A, EGF and pentraxin domain-containing protein 1	rs687621	9	135637065	136637065	H4 (80.8%)	Yes
Hypertension	SeqId_16751_15	NPPB	Natriuretic peptides B	rs198379	1	11415467	12415467	H3 (~100%)	No
Hypertension	SeqId_16751_15	NPPB	Natriuretic peptides B	rs198375	1	11413757	12413757	H3 (~100%)	No
Hypertension	SeqId_4297_62	SPON1	Spondin-1	rs601338	19	48706674	49706674	H4 (97.2%)	Yes
Hypertension	SeqId_18289_16	CCL15	C-C motif chemokine 15	rs681343	19	48706462	49706462	H4 (97.6%)	Yes
Hypertension	SeqId_18289_16	CCL15	C-C motif chemokine 15	rs492602	19	48706417	49706417	H4 (97.2%)	Yes
Hypertension	SeqId_7145_1	ITIH3	Inter-alpha-trypsin inhibitor heavy chain H3	rs2535629	3	52333219	53333219	H3 (61.0%)	No
A Fib	SeqId_11109_56	SVEP1	Sushi, von Willebrand factor type A, EGF and pentraxin domain-containing protein 1	rs4544697	1	154534632	155534632	H3 (~100%)	No
A Fib	SeqId_4297_62	SPON1	Spondin-1	rs10832169	11	13566486	14566486	H4 (92.8%)	Yes
A Fib	SeqId_4297_62	SPON1	Spondin-1	rs1969539	11	13538621	14538621	H4 (90.7%)	Yes
Diabetes	SeqId_3438_10	FSTL3	Follistatin-related protein 3	rs1260326	2	27230940	28230940	H4 (99.8%)	Yes
Diabetes	SeqId_5542_22	NRP1	Neuropilin-1	rs4665972	2	27098097	28098097	H4 (99.5%)	Yes
Diabetes	SeqId_5542_22	NRP1	Neuropilin-1	rs1260326	2	27230940	28230940	H4 (99.1%)	Yes
CHD	SeqId_18289_16	CCL15	C-C motif chemokine 15	rs28929474	14	94344947	95344947	H4 (99.8%)	Yes
CHD	SeqId_7145_1	ITIH3	Inter-alpha-trypsin inhibitor heavy chain H3	rs2535629	3	52333219	53333219	H3 (80.7%)	No
CHD	SeqId_5542_22	NRP1	Neuropilin-1	rs2506150	10	32983308	33983308	H4 (56.3%)	No
CHD	SeqId_5542_22	NRP1	Neuropilin-1	rs2506149	10	32980713	33980713	H4 (73.6%)	No
CHD	SeqId_5542_22	NRP1	Neuropilin-1	rs4846913	1	229794715	230794715	H4 (84.3%)	Yes
CHD	SeqId_5542_22	NRP1	Neuropilin-1	rs10864728	1	229804914	230804914	H4 (90.5%)	Yes
LVEDV	SeqId_3438_10	FSTL3	Follistatin-related protein 3	rs1260326	2	27230940	28230940	H4 (62.9%)	No
LVEDV	SeqId_10391_1	ANGPTL	Angiotensin-related protein 3	rs74076477	1	62266601	63266601	H1 (99.8%)	No
LVEDV	SeqId_18289_16	CCL15	C-C motif chemokine 15	rs28929474	14	94344947	95344947	H4 (99.6%)	Yes
LVEDV	SeqId_5542_22	NRP1	Neuropilin-1	rs4665972	2	27098097	28098097	H1 (58.4%)	No
LVEDV	SeqId_5542_22	NRP1	Neuropilin-1	rs1260326	2	27230940	28230940	H1 (50.3%)	No
LVEDV	SeqId_4297_62	SPON1	Spondin-1	rs10832169	11	13566486	14566486	H4 (99.8%, with SuSiE)	Yes
LVEDV	SeqId_4297_62	SPON1	Spondin-1	rs1969539	11	13538621	14538621	H1 (53.6%)	No
LVESV	SeqId_3320_49	IGFBP7	Insulin-like growth factor-binding protein 7	rs9271147	6	32077385	33077385	H3 (87.9%)	No
LVESV	SeqId_18289_16	CCL15	C-C motif chemokine 15	rs28929474	14	94344947	95344947	H4 (99.4%)	Yes
LVESV	SeqId_4297_62	SPON1	Spondin-1	rs10832169	11	13566486	14566486	H4 (97.3%)	Yes
LVESV	SeqId_4297_62	SPON1	Spondin-1	rs1969539	11	13538621	14538621	H4 (97.3%)	Yes
LVEF	SeqId_3320_49	IGFBP7	Insulin-like growth factor-binding protein 7	rs9271147	6	32077385	33077385	H3 (76.9%)	No
LVEF	SeqId_5636_10	MFAP4	Microfibril-associated glycoprotein 4	rs139356332	17	18789286	19789286	H1 (71.3%)	No
LVEF	SeqId_4297_62	SPON1	Spondin-1	rs10832169	11	13566486	14566486	H4 (95.0%)	Yes
LVEF	SeqId_4297_62	SPON1	Spondin-1	rs1969539	11	13538621	14538621	H4 (94.9%)	Yes

2. While it is true that the winner’s curse may result in larger effect size, a larger effect size is less likely to be obtained by chance, which implies a potential impact on statistical significance. I think that the authors should add this potential limitation to the Discussion section.

As suggested by the Reviewer, we have now added the following statement to the Limitations paragraph of the Discussion section:

Discussion Section, final paragraph, page 33:

“Two-sample MR used GWAS data from HERMES for the HF outcome and UKBB for the CMR outcomes. As some individuals in HERMES and UKBB may have contributed to pQTL data, there is a risk of winner’s curse which may result in larger effect size and potentially impact statistical significance.”

Reviewers' Comments:

Reviewer #1:

Remarks to the Author:

Thank you for providing an updated version of your manuscript that addresses my previous concerns. I have no further comments.